# Any-stepsize Gradient Descent for Separable Data under Fenchel–Young Losses

**Han Bao**[*]
The Institute of Statistical Mathematics
bao.han@ism.ac.jp

**Shinsaku Sakaue**[*]
CyberAgent
shinsaku.sakaue@gmail.com

**Yuki Takezawa**
Kyoto University and OIST
takezawa@ml.ist.i.kyoto-u.ac.jp

## Abstract

The gradient descent (GD) has been one of the most common optimizer in machine learning. In particular, the loss landscape of a neural network is typically sharpened during the initial phase of training, making the training dynamics hover on the edge of stability. This is beyond our standard understanding of GD convergence in the stable regime where stepsize is chosen sufficiently smaller. Recently, Wu et al. [63] have shown that GD converges with much larger stepsize under linearly separable logistic regression. Although their analysis hinges on the self-bounding property of the logistic loss, which seems to be a cornerstone to establish a modified descent lemma, our pilot study shows that other loss functions without the self-bounding property can make GD attain arbitrarily small loss with large stepsize. To further understand what property of a loss function matters in GD, we aim to show large-stepsize GD convergence for a general loss function based on the framework of *Fenchel–Young losses*. We essentially leverage the classical perceptron argument to derive the iteration complexity for achieving $\varepsilon$-optimal loss, which is possible for a majority of Fenchel–Young losses. This convergence result highlights that the self-bounding property may not be necessary for GD to attain arbitrarily small loss. Moreover, when a loss function entails *separation margin*, a notion relevant to the margin in support vector machines, GD often yields faster convergence than typical GD rate $T = \Omega(\varepsilon^{-1})$ for convex smooth objectives. Specifically, GD with the Tsallis entropy attains $\varepsilon$-optimal loss with the rate $T = \Omega(\varepsilon^{-1/2})$, and the Rényi entropy achieves the far better rate $T = \Omega(\varepsilon^{-1/3})$.

## 1 Introduction

Gradient-based optimizers are prevalent in the modern machine learning community with deep learning thanks to its scalability and plasticity. Among many variants, GD remains to be a standard choice. GD with constant stepsize is written as follows:

$$\mathbf{w}_{t+1} \coloneqq \mathbf{w}_t - \eta \nabla L(\mathbf{w}_t), \qquad \text{for } t = 0, 1, \ldots, T-1, \tag{GD}$$

where $\mathbf{w} \in \mathbb{R}^d$ is the optimization variables, $L(\cdot)$ is the loss function, and $\eta > 0$ is stepsize fixed across all steps. The *descent lemma* [42, Section 1.2.3] is a key to GD convergence: for $\beta$-smooth objective $L$, the stepsize choice $\eta < 2/\beta$ ensures that $L(\mathbf{w}_t)$ monotonically decreases. Nonetheless, little optimization theory has been known beyond the threshold $\eta > 2/\beta$; though modern neural networks exhibit much smaller smoothness values than practically used stepsize values [66, 57].

---

[*]This work was primarily conducted during the period when HB was affiliated with Kyoto University, and SS with the University of Tokyo and RIKEN AIP.

39th Conference on Neural Information Processing Systems (NeurIPS 2025).

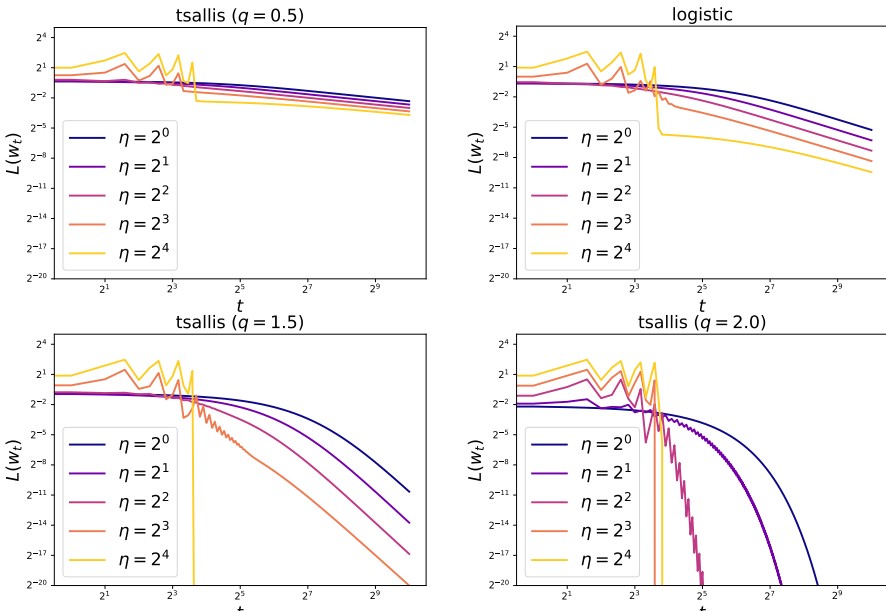

**Figure 1:** Pilot studies of GD with the same toy dataset as [63]. The dataset consists of four points, $\mathbf{x}_1 = [1, 0.2]^\top$, $y_1 = 1$, $\mathbf{x}_2 = [-2, 0.2]^\top$, $y_2 = 1$, $\mathbf{x}_3 = [-1, -0.2]^\top$, $y_3 = -1$, $\mathbf{x}_4 = [2, -0.2]^\top$, and $y_4 = -1$. GD is run with initialization $\mathbf{w}_0 = [0, 0]^\top$. Note that the logistic loss corresponds to the Tsallis 1-loss. The Tsallis 2- and $q$-loss are also known as the modified Huber loss [67] and $q$-entmax loss [45], respectively.

Moreover, recent studies have reported that GD trajectories of neural networks tend to inflate the sharpness of the loss landscape and hover on the *edge of stability* (EoS) before convergence [34, 18, 2].

Among several recent developments in the theory of large-stepsize GD (which we will review in Section 1.1), Wu et al. [63] investigated the large-stepsize behavior of GD by using the binary logistic regression with a linearly separable data, a minimal synthetic setting. They showed that GD initially oscillates with non-monotonic loss values (the EoS phase), which terminates in finite time (phase transition), and then the loss value decreases monotonically (the stable phase). Beyond the logistic loss, these results have been extended to loss functions with the *self-bounding property*: for a differentiable loss function $\ell \colon \mathbb{R} \to \mathbb{R}$ and its absolute derivative $g(\cdot) \coloneqq |\ell'(\cdot)|$, $\ell$ satisfies

$$\ell(z) \le \ell(x) + \ell'(x)(z - x) + C_\beta g(x)(z - x)^2 \quad \forall z, x \text{ with } |z - x| < 1, \quad \text{for some } C_\beta > 0. \quad (1)$$

The self-bounding property generalizes the polynomially-tailed loss [31, 30], and refines the standard smoothness property by allowing the smoothness modulus locally adaptive to the derivative, such that $C_\beta g(x)$. Thus, large $\eta$ can be cancelled out with the vanishingly small loss gradient after the phase transition [63, Lemma 29], and GD follows the descent direction.

In this paper, we study GD with large stepsize under a wide range of loss functions to identify a key factor to induce the convergent behavior. This is motivated by our pilot study shown in Figure 1, where we found that GD with large stepsize such as $\eta = 2^4$ remains to converge under the Tsallis $q$-loss (detailed in Section 4), even if the stepsize has gone beyond the classical stable regime. It is noteworthy therein that the Tsallis $q$-loss with $q > 1$ does not enjoy the self-bounding property. How much does the self-bounding property play a vital role in large-stepsize GD convergence? We specifically consider *Fenchel–Young losses* [11], a class of convex loss functions generated by a potential function $\phi$, as a template of loss functions. Fenchel–Young losses have been used in applications such as structured prediction [43], differentiable programming [10], and model selection [7], while being used as a theoretical tool for online learning [51, 52]. We identify that Fenchel–Young losses with *separation margin* (formally introduced in Section 2), a relevant notion to the margin in support vector machines, can often benefit from better GD convergence rates. We say a loss function has separation margin if the loss value vanishes with a sufficiently large positive prediction margin. Specifically, our main result is informally stated as follows.

**Theorem 1** (Informal version of Theorem 5). *Consider a binary classification dataset that is linearly separable. We run* (GD) *with arbitrary constant stepsize $\eta > 0$ and initialization $\mathbf{w}_0 = \mathbf{0}$ under*

*a Fenchel–Young loss generated by twice continuously differentiable and convex potential $\phi$ with separation margin. For $\varepsilon > 0$, after at most $T$ steps of* (GD)*, where*

$$T = \Omega(\varepsilon^{-\alpha}) \quad and \quad \alpha = \limsup_{\mu \downarrow 0} \frac{\phi'(\mu)}{\mu \phi''(\mu)} \left[ 1 - \frac{\phi(\mu)}{\mu \phi'(\mu)} \right],$$

*we have $L(\mathbf{w}_T) \leq \varepsilon$.*[1]

As defined in Section 2, a loss function with separation margin vanishes for a sufficiently large prediction margin, which is a natural indicator of correct classification used in support vector machines. The order of the convergence rate $T = \Omega(\varepsilon^{-\alpha})$ differs across various potential $\phi$. With a specific choice, the rate can be $T = \Omega(\varepsilon^{-1/2})$ (with $\phi$ being the Tsallis 2-entropy) and $T = \Omega(\varepsilon^{-1/3})$ (with $\phi$ being the Rényi 2-entropy, also known as the collision entropy [14]). Remarkably, these convergence rates are better than the classical GD convergence rate $T = \Omega(\varepsilon^{-1})$ under the stable regime, and even better than the convergence rate of the logistic loss after undergoing the EoS and phase transition [63]. Both the Tsallis and Rényi entropies above lack the self-bounding property but have separation margin. Therefore, we advocate the importance of separation margin for better GD convergence rates. We compare different Fenchel–Young losses in Section 4 and contrast our convergence result with the EoS and implicit bias in Section 5.

We present Theorem 1 formally in Section 3. Our proof leverages the classical perceptron argument [44] without relying on the descent lemma at all. Intuitively speaking, we track the growth of the parameter alignment $\langle \mathbf{w}_t, \mathbf{w}_* \rangle$ with the optimal separator $\mathbf{w}_*$. When a loss entails separation margin, $\langle \mathbf{w}_t, \mathbf{w}_* \rangle$ cannot grow arbitrarily large (as we simulate in Figure 2 later) while each step of (GD) improves a lower bound on $\langle \mathbf{w}_t, \mathbf{w}_* \rangle$, leading to the convergence. Section 3.1 describes this proof overview in detail. This is different from the proof of Wu et al. [63], whose core is the modified decent lemma (recapped in Lemma 21 in the appendix) based on the self-bounding property. Although the perceptron argument is partially used therein [63], the average loss is finally controlled by the modified descent lemma, and thus the proof is only applicable to the self-bounding losses.

## 1.1 Related work

Gradient descent with large stepsize has attracted attention recently. Specifically, non-monotonic behaviors of loss functions [65] and the sharpness adaptivity to loss landscapes [34, 18] have been observed empirically. It was argued that the sharpness tends to initially increases until the classical stable regime breaks down, and hovers on this boundary, termed as the edge of stability [18]. This observation mainly sparks two questions: why the loss landscape hovers on the EoS, and why converging. Answering either question must go beyond the classical optimization theory under the stable regime.

On why hovering on the EoS, let us make a brief review, though it is not a central focus of this paper: Ahn et al. [2] is a seminal work to empirically investigate the homogeneity of loss functions contributes to maintain the EoS. Later, it was showed that normalized GD (represented by scale-invariant losses) adaptively leads their intrinsic stepsize toward sharpness reduction [36]. The sharpness fluctuation is often attributed to the non-negligible third-order Taylor remainder of the loss landscape [37, 20].

We rather focus on why GD attains arbitrarily small loss with much larger stepsize. In this line, previous studies show convergence based on specific models such as multi-scale loss function [32], quadratic functions [5], matrix factorization [59, 17], a scalar multiplicative model [68, 33], a sparse coding model [3], and linear logistic regression [62]. Among them, we advocate the logistic regression setup proposed by Wu et al. [62] because it is relevant to implicit bias of GD [54, 29, 46], and moreover, Wu et al. [63] corroborates the benefit of large stepsize in GD convergence rate. Our work is provoked by Wu et al. [63], questioning what structure in a loss function leads GD to arbitrarily small loss. Indeed, we observe in Figure 1 that loss functions without the self-bounding property (1) can make GD attain arbitrarily small loss, though the self-bounding property seems essential to calm the EoS down to the stable phase [63] as well as to establish the max-margin directional convergence [29, 46]. A similar question to ours is raised by Tyurin [58], who argues that the stable convergence of large-stepsize logistic regression might be an artifact due to the functional

---

[1]Throughout this paper, we consider $\eta = \Theta(1)$ with respect to the error tolerance $\varepsilon$ when we say arbitrary stepsize unless otherwise noted.

form of the logistic loss—eventually Tyurin [58] argued that large-stepsize logistic regression behaves like the classical perceptron. To this end, we show in Theorem 5 that arbitrary-stepsize GD can converge under a wide range of losses even without the self-bounding property (1), and moreover, occasionally yielding a better rate than the classical stable convergence rate. We discuss it more in Section 5. Note that one work attempts to extend the separable logistic regression setup to the non-separable one [41]; yet, we still do not have satisfactory results beyond the one-dimensional case. Due to its intricateness, we follow the separable case.

Lastly, our work benefits the study of regret bounds of surrogate losses [9, 1, 23, 6, 38, 8]. A surrogate regret bound connect a surrogate loss to a downstream task loss, while the optimization error of the surrogate loss is usually ignored. Our GD convergence analysis can be integrated to surrogate regret bounds when discussing a downstream task performance.

## 1.2 Notation

Let $\mathbb{R}_{\geq 0}$ be the set of nonnegative reals. Let $[n] := \{1, \ldots, n\}$ for $n \in \mathbb{N}$. Let $\mathbf{1}$ be the all-ones vector and $\mathbf{e}_i \in \mathbb{R}^d$ be the $i$-th standard basis vector, i.e., all zeros except for the $i$-th entry being one. For $\mathcal{S} \subseteq \mathbb{R}^d$, $\mathrm{int}(\mathcal{S})$ denotes its (relative) interior, and $I_{\mathcal{S}} \colon \mathbb{R} \to \{0, \infty\}$ its indicator function, which takes zero if $\boldsymbol{\mu} \in \mathcal{S}$ and $\infty$ otherwise. For $\Omega \colon \mathbb{R}^d \to \mathbb{R} \cup \{\infty\}$, $\mathrm{dom}(\Omega) := \{ \boldsymbol{\mu} \in \mathbb{R}^d \mid \Omega(\boldsymbol{\mu}) < \infty \}$ denotes its effective domain and $\Omega^*(\boldsymbol{\theta}) := \sup \{ \langle \boldsymbol{\theta}, \boldsymbol{\mu} \rangle - \Omega(\boldsymbol{\mu}) \mid \boldsymbol{\mu} \in \mathbb{R}^d \}$ its convex conjugate. Let $\triangle^d := \{ \boldsymbol{\mu} \in \mathbb{R}^d_{\geq 0} \mid \langle \mathbf{1}, \boldsymbol{\mu} \rangle = 1 \}$ be the probability simplex. We introduce $\mathcal{C}^k(\mathcal{I})$ as the set of $k$-th continuously differentiable functions on the interval $\mathcal{I} \subseteq \mathbb{R}$.

Let $\Psi \colon \mathbb{R}^d \to \mathbb{R} \cup \{\infty\}$ be a strictly convex function differentiable throughout $\mathrm{int}(\mathrm{dom}\,\Psi) \neq \varnothing$. We say $\Psi$ is of *Legendre-type* if $\lim_{i \to \infty} \|\nabla \Psi(\mathbf{x}_i)\|_2 = \infty$ whenever $\mathbf{x}_1, \mathbf{x}_2, \ldots$ is a sequence in $\mathrm{int}(\mathrm{dom}\,\Psi)$ converging to a boundary point of $\mathrm{int}(\mathrm{dom}\,\Psi)$ (see [49, Section 26]).

## 2 Preliminary on Fenchel–Young losses

Fenchel–Young losses have been introduced by Blondel et al. [11] as a general class of surrogate losses for structured prediction, which are classification-calibrated [60]. This can be seen as a Bregman divergence comparing primal and dual points [25, 4]. Despite that the logistic and hinge losses are widely prevailing in practice, we can improve the performance of some prediction tasks by changing specific Fenchel–Young losses, as reported by Roulet et al. [50]. We choose Fenchel–Young losses because a vast majority of convex, Lipschitz, and classification-calibrated losses are included in this class—otherwise, GD convergence is hardly obtained beyond the edge of stability. Moreover, the separation margin property, one of the key features of Fenchel–Young losses, controls GD behaviors significantly.

**Definition 2.** *Let* $\Omega \colon \mathbb{R}^K \to \mathbb{R} \cup \{\infty\}$ *be a potential function. The* Fenchel–Young *loss* $\ell_\Omega \colon \mathrm{dom}(\Omega^*) \times \mathrm{dom}(\Omega) \to \mathbb{R}_{\geq 0}$ *generated by* $\Omega$ *is defined as*

$$\ell_\Omega(\boldsymbol{\theta}; \boldsymbol{\mu}) := \Omega^*(\boldsymbol{\theta}) + \Omega(\boldsymbol{\mu}) - \langle \boldsymbol{\theta}, \boldsymbol{\mu} \rangle \,.$$

In multiclass classification, $\ell_\Omega(\boldsymbol{\theta}; \boldsymbol{\mu})$ measures the proximity between a score $\boldsymbol{\theta}$ and a target label $\boldsymbol{\mu} = \mathbf{e}_i$ (for a class $i \in [K]$). By definition, $\ell_\Omega(\cdot, \boldsymbol{\mu})$ is convex for any $\boldsymbol{\mu} \in \mathrm{dom}(\Omega)$. Moreover, $\ell_\Omega(\boldsymbol{\theta}; \boldsymbol{\mu}) = 0$ holds if and only if $\boldsymbol{\mu} \in \partial\Omega^*(\boldsymbol{\theta})$ due to the equality condition of the Fenchel–Young inequality.

We follow [11, Section 4.4] to consider binary ($K = 2$) loss functions. The following set of assumptions is imposed on a potential function $\Omega$. The asymmetric generalization is possible, but we choose to keep the analysis simpler so that we can focus more on the essence of GD convergence.

**Assumption 1.** *For a potential function* $\Omega$, *assume* $\mathrm{dom}(\Omega) \subseteq \triangle^K$ *and that* $\Omega$ *satisfies the zero-entropy condition* $\Omega(\boldsymbol{\mu}) = 0$ *for* $\boldsymbol{\mu} \in \{\mathbf{e}_i\}_{i \in [K]}$; *convexity* $\Omega((1 - \alpha)\boldsymbol{\mu} + \alpha\boldsymbol{\mu}') \leq (1 - \alpha)\Omega(\boldsymbol{\mu}) + \alpha\Omega(\boldsymbol{\mu}')$ *for* $\boldsymbol{\mu} \neq \boldsymbol{\mu}'$ *and* $\alpha \in (0, 1)$; *symmetry* $\Omega(\boldsymbol{\mu}) = \Omega(\mathbf{P}\boldsymbol{\mu})$ *for any* $K \times K$ *permutation* $\mathbf{P}$.

Let us restrict ourselves to $K = 2$ (binary classification) and write $\phi(\mu) := \Omega([\mu, 1 - \mu]^\top)$. If we choose $\boldsymbol{\theta} = [s, -s]^\top \in \mathbb{R}^2$ as a score vector, the Fenchel–Young loss can be written as

$$\ell_\Omega(\boldsymbol{\theta}; \mathbf{e}_i) = \begin{cases} \phi^*(-s) & \text{if } i = 1, \\ \phi^*(s) & \text{if } i = 2, \end{cases}$$

and $\operatorname{dom}(\phi^*) = \mathbb{R}$. Hence, the Fenchel–Young loss is simplified as $\phi^*(-ys)$ if we relabel two classes $i = 1$ and $i = 2$ with $y = 1$ and $y = -1$, respectively. Thus, we suppose the form of a symmetric margin-based loss function $\ell(z) := \phi^*(-z)$. Therein, a Fenchel–Young loss $\ell$ extends a proper canonical composite loss [47] over the entire prediction space $z \in \mathbb{R}$, as discussed in [7].

**Separation margin.** For specific potential functions, Fenchel–Young losses entail *separation margin* [11, Section 5], which is a generalized notion of classical margin in support vector machines.

**Definition 3.** *For a loss $\ell \colon \mathbb{R} \to \mathbb{R}_{\geq 0}$, we say $\ell$ has the* separation margin property *if there exists $m > 0$ such that any prediction $z \geq m$ incurs $\ell(z) = 0$. The smallest $m$ is the* separation margin *of $\ell$.*

Hence, $\ell(z) = 0$ indicates the prediction $z \in \mathbb{R}$ not only correctly classifies a given point but also has safe margin $m$ away from the classification boundary $z = 0$. It is shown that the existence of the separation margin property can be tested through the subgradient $\partial \phi$ [11, Proposition 6].

**Proposition 4** ([11]). *A Fenchel–Young loss $\ell(z) = \phi^*(-z)$ satisfying Assumption 1 has separation margin if and only if $\partial \phi(\mu) \neq \varnothing$ for any $\mu \in [0, 1]$. When $\phi \in \mathcal{C}^1((0, 1))$ has separation margin $m$,*

$$m = -\lim_{\mu \downarrow 0} \phi'(\mu).$$

For a differentiable $\phi$, the nonempty-subgradient condition requires that the derivative $\phi'(\mu)$ does not explode at the boundary points of the domain $\mu \in \operatorname{dom}(\phi) = \{0, 1\}$. In this case, $\phi$ is *not* of Legendre-type [49]. As we will see later, the convergence behavior of GD hinges on the separation margin property of a loss function. More detailed analysis of the separation margin property for binary classification can be found in [7]. In Section E, we show that a loss satisfying the self-bounding inequality (1) does not have separation margin (but not the other way around).

**Examples.** With the Shannon negentropy $\phi(\mu) = \mu \ln \mu + (1 - \mu) \ln(1 - \mu)$, we recover the logistic loss $\phi^*(-z) = \ln(1 + \exp(-z))$. With the negative of the Gini index $\phi(\mu) = \mu^2 - \mu$, we can generate the modified Huber loss $\phi^*(-z) = \max\{0, 1 - z\}^2 / 4$ if $z \geq -1$ and $\phi^*(-z) = -z$ otherwise [67], which is the binarized sparsemax loss [39]. If we choose $\phi(\mu) = \max\{\mu, 1 - \mu\}$, we recover the hinge loss $\phi^*(-z) = \max\{0, 1 - z\}$. We discuss more examples in Section 4.

# 3 Convergence of large stepsize GD under Fenchel–Young losses

We consistently assume that the dataset is bounded and linearly separable.

**Assumption 2.** *Assume the training data $(\mathbf{x}_i, y_i)_{i \in [n]}$ satisfies*

- *for every $i \in [n]$, $\|\mathbf{x}_i\| \leq 1$ and $y_i \in \{\pm 1\}$;*
- *there is $\gamma > 0$ and a unit vector $\mathbf{w}_*$ such that $\langle \mathbf{w}_*, \mathbf{z}_i \rangle \geq \gamma$ for every $i \in [n]$, where $\mathbf{z}_i := y_i \mathbf{x}_i$.*

Instead of logistic regression, we choose a Fenchel–Young loss $\ell(z) = \phi^*(-z)$ associated with a binary potential function $\phi$, and minimize the following risk by (GD) with fixed stepsize $\eta > 0$ to learn a linear classifier $\mathbf{w}$:[2]

$$L(\mathbf{w}) := \frac{1}{n} \sum_{i \in [n]} \ell(\langle \mathbf{w}, y_i \mathbf{x}_i \rangle) = \frac{1}{n} \sum_{i \in [n]} \ell(\langle \mathbf{w}, \mathbf{z}_i \rangle). \tag{2}$$

We impose the following assumptions on our loss function.

**Assumption 3.** *Consider a loss $\ell \colon \mathbb{R} \to \mathbb{R}_{\geq 0}$.*

A. **Fenchel–Young loss.** *Assume that $\ell(z)$ is a Fenchel–Young loss $\phi^*(-z)$ generated by a potential $\phi : \mathbb{R} \to \mathbb{R} \cup \{\infty\}$ such that $\phi \in \mathcal{C}^2((0, 1))$ satisfies Assumption 1, $\phi$ is strictly convex, and $\phi'' > 0$ on the interval $(0, 1)$.*

B. **Regularity.** *Assume that $\rho(\lambda) := \min_{z \in \mathbb{R}} \lambda \ell(z) + z^2$ (for $\lambda \geq 1$) is well-defined.*

---

[2]To generalize the linear model, one straightforward way is to focus on deep homogeneous networks [56]. We stay on the linear model for now because the straightforward extension to deep homogeneous networks may not significantly change the problem structure.

**C. Lipschitz continuity.** For $g(\cdot) := |\ell'(\cdot)|$, assume $g(\cdot) \le C_g$ for some $C_g > 0$.

We will later see that $\rho$ characterizes the growth rate of the parameter norm $\|\mathbf{w}_t\|$ during GD in (5). This notion is inherited from [63]. Now, we are ready to state our main result, the GD rate to attain arbitrarily small loss for linearly separable data under Fenchel–Young losses. Remarkably, we show convergence without the self-bounding property of a loss function, unlike [63].

**Theorem 5** (Main result)**.** *Suppose Assumption 2 and consider* (GD) *with stepsize $\eta > 0$ and $\mathbf{w}_0 = \mathbf{0}$ under a Fenchel–Young loss $\ell$ satisfying Assumption 3. For any $\bar{\varepsilon} \in (0, 1)$, let*

$$\alpha := \sup_{\mu \in (0, \bar{\varepsilon}]} \frac{\phi'(\mu)}{\mu \phi''(\mu)} \left[ 1 - \frac{\phi(\mu)}{\mu \phi'(\mu)} \right] \quad and \quad C_\phi := \frac{\bar{\mu}}{[\bar{\mu}\phi'(\bar{\mu}) - \phi(\bar{\mu})]^\alpha}, \tag{3}$$

*where $C_\phi > 0$ depends on $\phi$ and $\bar{\varepsilon}$ solely and $\bar{\mu} := \min\left\{ g(\ell^{-1}(\bar{\varepsilon})), 1 \right\}$. If $\alpha, C_\phi \in (0, \infty)$ and*

$$for\ \varepsilon \in (0, \bar{\varepsilon}), \quad T > \frac{n}{C_\phi \gamma^2} \left( \frac{4\sqrt{\rho(\gamma^2 \eta T)}}{\eta} + C_g \right) \varepsilon^{-\alpha}$$

*holds, then we have $\min_{t \in [T]} L(\mathbf{w}_t) \le \varepsilon$.*

This convergence guarantee even applies to non-smooth Fenchel–Young losses as long as Assumption 3 is satisfied—note that $\phi$ must be strongly convex to ensure the smoothness of the associated Fenchel–Young loss [11, Proposition 2.4]. As seen later in Section 4, $\alpha$ and $C_\phi$ neither diverge nor degenerate for arbitrarily small $\bar{\varepsilon}$ under many examples of $\phi$. When $\ell$ has separation margin, Lemma 9 in Section A provides a finite upper bound on $\rho$, yielding the following simpler form.

**Corollary 6.** *Under the same setup with Theorem 5, we additionally assume that $\ell$ has separation margin $m > 0$. For any $\bar{\varepsilon} \in (0, 1)$, if $(\alpha, C_\phi)$ defined in (3) satisfies $\alpha, C_\phi \in (0, \infty)$ and*

$$for\ \varepsilon \in (0, \bar{\varepsilon}), \quad T > \frac{n}{C_\phi \gamma^2} \left( \frac{4m}{\eta} + C_g \right) \varepsilon^{-\alpha}$$

*holds, then we have $\min_{t \in [T]} L(\mathbf{w}_t) \le \varepsilon$.*

A loss function without separation margin does not have finite $\rho$, which typically yields slower convergence as we see in Section 4. Therefore, (GD) operated on many common Fenchel–Young losses converges under the separability, regardless of the choice of $\eta$. Note that the classical GD convergence analysis under convex smooth functions provides $T = \Omega(\varepsilon^{-1})$. As we see later in Section 4, some loss functions entail better rates with $\alpha < 1$, summarized in Table 1.

## 3.1 Proof outline

The proof of Theorem 5 essentially relies on the perceptron convergence analysis [44] and the asymptotical order evaluation of rate functions [6]. We sketch the proof in this section to highlight the structure of the GD convergence in our setup and complete the proof in Section B.

When we show the convergence of perceptron, we leverage an inequality of the following type:

$$\underbrace{C_{\mathrm{L}} t \le \langle \mathbf{w}_t, \mathbf{w}_* \rangle}_{(\clubsuit)} \le \underbrace{\|\mathbf{w}_t\| \le C_{\mathrm{U}}(t)}_{(\diamond)} \quad \text{for } t \ge 1, \tag{4}$$

where $C_{\mathrm{L}} > 0$ is a non-degenerate constant independent of $t$. The inequality $(\clubsuit)$ holds only while perceptron misclassifies some examples. Thus, perceptron correctly classifies all examples after at most $T$ iterations such that $C_{\mathrm{L}} T > C_{\mathrm{U}}(T)$. Such $T$ exists as long as $C_{\mathrm{U}}(t)$ is sublinear in $T$.

When it comes to our setup, an inequality $(\clubsuit)$ is obtained by recursively expanding the update (GD)[3]

$$\langle \mathbf{w}_t, \mathbf{w}_* \rangle \ge \langle \mathbf{w}_{t-1}, \mathbf{w}_* \rangle + \frac{\gamma \eta}{n} g(\langle \mathbf{w}_{t-1}, \mathbf{z}_{i_{t-1}} \rangle) \ge \cdots \ge \frac{\gamma \eta}{n} \sum_{k=0}^{t-1} g(\langle \mathbf{w}_k, \mathbf{z}_{i_k} \rangle) =: \gamma \eta \widetilde{G}(\mathbf{w}_k),$$

---

[3]This expansion relies on $\langle \mathbf{w}_*, \mathbf{z}_{i_{t-1}} \rangle \ge \gamma$, namely, the linear separability in Assumption 2. To lift the linear separability assumption, we may require to introduce additional distributional assumptions here.

where $\mathbf{z}_{i_k}$ is a misclassified example by $\mathbf{w}_k$. Perceptron enjoys an inequality of (♣)-type immediately because it optimizes the loss function $\ell_{\mathrm{per}}(z) = \max\{-z, 0\}$, which yields $g(z) = 1$ if $z < 0$ (i.e., if misclassified). When considering a Fenchel–Young loss satisfying Assumption 3, we do not have a non-degenerate lower bound for $g(z)$ because we can make $g(z)$ arbitrarily close to zero. Instead, we lower-bound $g(z)$ by a (non-degenerate) error tolerance $\varepsilon_1 > 0$, $g(z) \geq \varepsilon_1$, before we attain the $\varepsilon$-optimal loss. Lemma 11 and (a part of) Lemma 13 in Section B are relevant to (♣). Note that the perceptron argument is used in [63] but in a different way: they control $L(\mathbf{w}_k)$ through the upper bound on $\widetilde{G}(\mathbf{w}_k)$ (see Lemma 25). This is applicable only to self-bounding losses.

To obtain an inequality of (◇)-type, by following the standard perceptron analysis, we directly expand the update (GD) $\|\mathbf{w}_t\|^2 = \|\mathbf{w}_{t-1} - \eta \nabla L(\mathbf{w}_{t-1})\|^2$ recursively, and upper-bound it by noting that $\ell_{\mathrm{per}}$ has separation margin, leading to $C_{\mathrm{U}}(t) = \mathcal{O}(\sqrt{t})$. Though this is possible for a Fenchel–Young loss with separation margin, we can improve this bound by borrowing the *split optimization technique*, introduced by [63]. Eventually, we can upper-bound $\|\mathbf{w}_t\|$ as follows:

$$\|\mathbf{w}_t\| \leq \frac{4\sqrt{\rho(\gamma^2 \eta t)} + \eta C_g}{\gamma}. \tag{5}$$

In particular, we have $\rho(\lambda) = \mathcal{O}(1)$ when a loss has separation margin (see Lemma 9), and therein $C_{\mathrm{U}}(t) = \mathcal{O}(1)$. This is where separation margin plays a crucial role. We recap the split optimization technique in Lemma 10, based on which Lemma 19 in Section D shows this inequality of (◇)-type.

The remaining piece is to assess the order of the convergence rate. After solving the inequality (♣, ◇) with $t = T$ being the stopping time, we have $T$ as a function of the error tolerance $\varepsilon$, $T = f(\varepsilon)$, where $f$ is a nondecreasing rate function depending on $\phi$. To characterize the asymptotic order at vanishing $\varepsilon$, we attempt to evaluate in the form $f(\varepsilon) \simeq \varepsilon^{\alpha_0}$ for an order parameter $\alpha_0 > 0$, which can be estimated by

$$\frac{\varepsilon f'(\varepsilon)}{f(\varepsilon)} \xrightarrow{\varepsilon \downarrow 0} \alpha_0, \quad \text{if the limit exists.}$$

Thus, the order parameter $\alpha_0$ is solely determined by the functional form of potential function $\phi$. This technique has been initially developed in functional analysis to estimate moduli of Banach and Orlicz spaces [53, 27, 13], and recently introduced in convex analysis to approximate a convex function by power functions [28] and estimate moduli of convexity [6, 8]. The general statement of the order evaluation is given in Lemma 12 and instantiated for GD convergence in Lemma 13 in Section B.

## 4    Examples of loss functions

Now, we instantiate Theorem 5 for several examples of Fenchel–Young losses to discuss the convergence rate. Instead of specifying a loss function $\ell(z) = \phi^*(-z)$, we directly specify its potential function $\phi$ subsequently. For each $\phi$, we compute $(\alpha, C_\phi)$ in (3) to investigate the convergence rate given by Theorem 5, by taking $\bar{\varepsilon}$ (and thus $\bar{\mu}$) vanishingly small. In addition, we can compute separation margin $m$ by Proposition 4 if exists; otherwise, we need to compute $\rho$ for a loss (see Lemma 27). Table 1 summarizes different loss functions and their GD convergence rates. All the detailed calculations are deferred to Section F, where we have an additional example of $\phi$ (pseudo-spherical entropy) with non-converging $\alpha$.

**Shannon entropy.**    Consider the binary Shannon (neg)entropy $\phi(\mu) = \mu \ln \mu + (1 - \mu) \ln(1 - \mu)$. The generated Fenchel–Young loss is the logistic loss $\ell(z) = \ln(1 + \exp(-z))$, which enjoys the self-bounding property and hence does not have separation margin (see Section E). The loss parameters are $\alpha = 1$ and $C_\phi = 1$. Moreover, we know $C_g = 1$ and $\rho(\lambda) \leq 1 + \ln^2(\lambda)$ [63]. Plugging this back to Theorem 5, we have the $\varepsilon$-optimal risk at most after

$$T \gtrsim \left[ \frac{4\sqrt{2}(\log_2(\gamma^2 \eta) + 1)}{\eta} + \frac{1}{\ln 2} \right] \frac{n\varepsilon^{-1}}{\gamma^2} \quad \text{iterations,}$$

where logarithmic factors in $\varepsilon^{-1}$ are ignored. This indicates the rate $T = \widetilde{\Omega}(\varepsilon^{-1})$, recovering the standard GD convergence rate under the stable regime but with arbitrary stepsize $\eta$. In Section 5, we compare this rate with [63] in more detail.

**Table 1:** Comparison of Fenchel–Young losses generated by different potential function $\phi$. Here, $m = \infty$ and $\beta = \infty$ indicate the lack of separation margin and smoothness, respectively. Since we do not have closed-form $\beta$ for the Rényi entropy with $q \in (1, 2)$, we merely show its lower bound. The convergence rates ignore the dependency on $\{m, n, \gamma, \eta\}$, and hold for arbitrary stepsize $\eta$ regardless of $\eta < 2/\beta$.

| Potential $\phi$ | Parameter $q$ | Sep. mgn. $m$ | Smoothness $\beta$ | Order $\alpha$ | Conv. rate for $T$ |
|---|---|---|---|---|---|
| Shannon | — | $\infty$ | $1/4$ | $1$ | $\widetilde{\Omega}(\varepsilon^{-1})$ |
| Semi-circle | — | $\infty$ | $1/4$ | $2$ | $\Omega(\varepsilon^{-4})$ |
| Tsallis | $(0,1)$ | $\infty$ | $\dfrac{2^{q-3}}{q}$ | $\dfrac{1}{q}$ | $\Omega(\varepsilon^{-2/q})$ |
| | $(1,2]$ | $\dfrac{q}{q-1}$ | | | $\Omega(\varepsilon^{-1/q})$ |
| | $(2,\infty)$ | | $\infty$ | $1/2$ | $\Omega(\varepsilon^{-1/2})$ |
| Rényi ♣ | $(0,1)$ | $\infty$ | $1/4q$ | $\dfrac{1}{q}$ | $\Omega(\varepsilon^{-2/q})$ |
| | $(1,2)$ | $\dfrac{q}{q-1}$ | $(\geq 1/4q)$ | | $\Omega(\varepsilon^{-1/q})$ |
| | $2$ | | $\infty$ | $1/3$ | $\Omega(\varepsilon^{-1/3})$ |

**Semi-circle entropy.** Consider $\phi(\mu) = -2\sqrt{\mu(1-\mu)}$. The generated Fenchel–Young loss (we call the semi-circle loss) $\ell(z) = (-z + \sqrt{z^2+4})/2$ enjoys the self-bounding property and does not have separation margin since $\phi'(\mu) \to -\infty$ as $\mu \downarrow 0$ (see Section E). The semi-circle loss is relevant to the exponential/boosting loss $\ell_{\exp}(z) = \exp(-z)$, which has the semi-circle entropy as the Bayes risk [15, 1]. The loss parameters are $\alpha = 2$ and $C_\phi = 1$. Moreover, we have $C_g = 1$ and $\rho(\lambda) \leq 5\lambda/(2\ln\lambda)$. Plugging this back to Theorem 5, we have the $\varepsilon$-optimal risk at most after

$$T > \underbrace{\frac{40n^6}{\gamma^2\eta\ln(2\gamma^2\eta)}\varepsilon^{-4}}_{\text{extra price for lacking separation margin}} + \frac{2n}{\gamma^2}\varepsilon^{-2} \quad \text{iterations,}$$

where the first term $\Omega(\varepsilon^{-4})$ is an extra price due to the lack of separation margin of the semi-circle loss. For arbitrary stepsize $\eta$, the convergence rate is $T = \Omega(\varepsilon^{-4})$, and stepsize $\eta$ as large as $\eta = \Omega(\varepsilon^{-2})$ improves the rate to be $T = \widetilde{\Omega}(\varepsilon^{-2})$ by cancelling the extra term out.

This convergence rate of the semi-circle loss is even worse than the GD convergence rate for general convex smooth functions, $T = \Omega(\varepsilon^{-1})$. This is because the perceptron argument is merely sufficient for GD convergence. Nonetheless, the perceptron argument more informatively states that we have $\langle \mathbf{w}_t, \mathbf{w}_* \rangle / \|\mathbf{w}_t\| \gtrsim \varepsilon^\alpha$ after minimizing the risk at the $\varepsilon$-optimal level—by combining the inequalities ($♣, \diamond$) (in Eq. (4)). This indicates that the loss function with larger $\alpha$ yields slower parameter alignment toward $\mathbf{w}_*$.

**Tsallis entropy.** For $q > 0$ with $q \neq 1$, consider the Tsallis $q$-(neg)entropy

$$\phi(\mu) = \frac{\mu^q + (1-\mu)^q - 1}{q-1}$$

generalizing the Shannon entropy for non-extensive systems [55]. It recovers the Shannon entropy at the limit $q \to 1$. The generated Fenchel–Young loss is known as the $q$-entmax loss [45]. We divide the case depending on parameter $q$:

- When $0 < q < 1$: $(\alpha, C_\phi) = (1/q, 1)$, and $\phi^*$ does not have separation margin.
- When $1 < q \leq 2$: $(\alpha, C_\phi) = (1/q, 1)$, and $\phi^*$ has separation margin $m = q/(q-1)$.
- When $2 < q$: $(\alpha, C_\phi) = (1/2, \sqrt{2/q})$, and $\phi^*$ has separation margin $m = q/(q-1)$.

For all cases, $\alpha$ and $C_\phi$ stay in $(0, \infty)$. The convergence rate is $T = \Omega(\varepsilon^{-2/q})$ for $q \in (0, 1)$ (by Corollary 28); $T = \Omega(\varepsilon^{-1/q})$ for $q \in (1, 2)$; $T = \Omega(\varepsilon^{-1/2})$ for $2 \leq q$. This suggests that we have a better convergence rate over the Shannon case when $q > 1$ and the best rate is $\Omega(\varepsilon^{-1/2})$.

**Rényi entropy.** For $q \in (0, 2] \setminus \{1\}$, consider the Rényi $q$-(neg)entropy

$$\phi(\mu) = \frac{1}{q - 1} \ln \left[ \mu^q + (1 - \mu)^q \right]$$

generalizing the Shannon entropy (with the limit $q \to 1$) while preserving additivity for independent events [48]. The Rényi entropy extended beyond $q > 2$ becomes nonconvex, which we do not consider. The Rényi 2-entropy is referred to as the collision entropy [14].

We divide the case depending on parameter $q$:

- When $0 < q < 1$: $(\alpha, C_\phi) = (1/q, 1)$, and $\phi^*$ does not have separation margin.
- When $1 < q < 2$: $(\alpha, C_\phi) = (1/q, 1)$, and $\phi^*$ has separation margin $m = q/(q - 1)$.
- When $q = 2$: $(\alpha, C_\phi) = (1/3, \sqrt[3]{3/8})$, and $\phi^*$ has separation margin $m = 2$.

For all cases, $\alpha$ and $C_\phi$ stay in $(0, \infty)$. The convergence rate is $T = \Omega(\varepsilon^{-2/q})$ for $q \in (0, 1)$ (by Corollary 28); $T = \Omega(\varepsilon^{-1/q})$ for $q \in (1, 2)$; $T = \Omega(\varepsilon^{-1/3})$ for $q = 2$. Surprisingly, we have a "leap" of the order from $\varepsilon^{-1/q}$ to $\varepsilon^{-1/3}$ as $q \uparrow 2$, and the convergence rate $\Omega(\varepsilon^{-1/3})$ is far better than the Shannon and Tsallis cases. When $q = 2$, Corollary 6 implies that we have the $\varepsilon$-optimal risk at most after

$$T > \sqrt[3]{8/3} \frac{n}{\gamma^2} \left( \frac{8}{\eta} + 1 \right) \varepsilon^{-1/3} \quad \text{iterations.}$$

## 5 Discussion and open problems

**Comparison with Wu et al. [63].** The large-stepsize logistic regression has been shown to exhibit the following phase transition [63]: the GD sequence initially stays in the EoS phase such that the risk $L(\mathbf{w}_t)$ fluctuates initially with its average $t^{-1} \sum_k L(\mathbf{w}_k)$ controlled. Once we experience $L(\mathbf{w}_t) \lesssim \min \{1/\eta, \ell(0)/n\}$, which is possible within $\mathcal{O}(\eta)$ steps at most, GD leaves the EoS and the loss converges in the rate $L(\mathbf{w}_t) = \widetilde{\mathcal{O}}(1/(\eta t))$. The stepsize $\eta$ trades off the phase transition time for the stable convergence rate. If we know the maximum number of steps $T$ in advance, the choice $\eta = \Theta(T)$ balances them, achieving the acceleration to $L(\mathbf{w}_t) = \widetilde{\mathcal{O}}(1/T^2)$. We detail them in Section D. This is arguably interesting to demonstrate how GD benefits from large stepsize.

Nevertheless, we would like to highlight two caveats. First, we must undergo $L(\mathbf{w}_t) \leq \ell(0)/n$ before exiting the EoS phase. This means that *our linear model has already classified all points correctly during the EoS phase* since any single point $\mathbf{z}_i$ incurs loss at most $\ell(\langle \mathbf{w}_s, \mathbf{z}_i \rangle) \leq \ell(0) \implies \langle \mathbf{w}_s, \mathbf{z}_i \rangle \geq 0$ (cf. Lemma 22 in Section D).[4] GD keeps improving the logistic loss after the stable phase just because the logistic loss does not enjoy separation margin and never touches strict zero. We refer interested readers to the relevant discussion in Tyurin [58], who argues that the faster convergence in the stable phase is attributed to the choice of the logistic loss.

Second, the EoS termination condition $L(\mathbf{w}_t) \lesssim 1/\eta$ suggests that the risk must be once $\mathcal{O}(1/T)$-optimal (with the optimally balancing choice $\eta = \Theta(T)$) before benefitting from the super-fast rate $\widetilde{\mathcal{O}}(1/T^2)$. Yet, GD under some loss functions including the Tsallis $q$-loss $(q > 1)$ and the Rényi $q$-loss $(q > 1)$ achieves better risk with the same GD steps. If our goal is simply to classify all training points, these alternative losses might do better jobs in terms of optimization solely.

**Self-bounding property and implicit bias.** Having said that, the phase transition may play an important role in implicit bias. It was shown under the linearly separable case that logistic regression optimized with GD enlarges the norm $\|\mathbf{w}_t\|$ toward the max-margin direction in rate $\Omega(\ln(t))$ [54, 62, 16]. Thus, we may argue that $\mathbf{w}_t$ gradually comes to classify all data points correctly during the EoS phase and evolves toward the max-margin direction in the stable phase.

We reported how $\|\mathbf{w}_t\|$ evolves under the pilot setup in Figure 2 with different loss functions. As seen, the logistic and Tsallis 0.5 losses inflate $\|\mathbf{w}_t\|$ endlessly, which do not have separation margin. In

---

[4]Cai et al. [16] extends Wu et al. [62] for two-layer near-homogeneous NNs, where it is not explicit that the model correctly classifies all points after the EoS phase. Taking a closer look, we can see that their Lemma A.7 leverages the well-controlled risk, which is an alternative expression to "$L(\mathbf{w}_s) \leq \ell(0)/n$" under their setup.

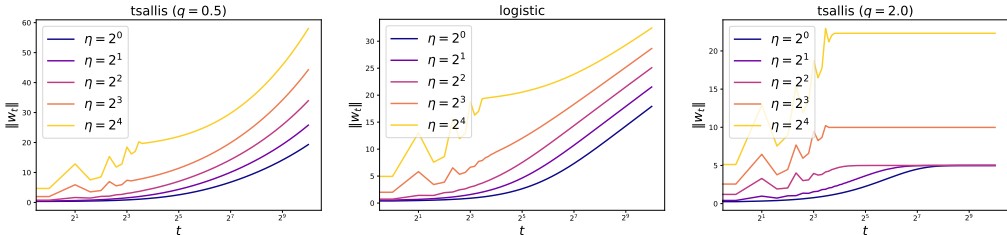

**Figure 2:** Under the same setup as Figure 1, we show $\|\mathbf{w}_t\|$ along the number of steps $t$ with different losses.

stark contrast, the Tsallis 2-loss prevents $\|\mathbf{w}_t\|$ from growing endlessly just because of its separation margin—recall the norm upper bounds of the norm $\|\mathbf{w}_t\|$ in (11) and the growth rate $\rho$ in Lemma 9. This raises two open questions: (1) Do we have similar implicit bias aligning toward the max-margin direction under a loss function with separation margin? (2) What are benefits and caveats of endless growing of $\|\mathbf{w}_t\|$? The latter is particularly relevant to the overconfidence issue due to excessively large $\|\mathbf{w}_t\|$ [61] and worse generalization due to prohibitively large within-class variance [26]. Wu et al. [64] argues that excessively large $\|\mathbf{w}_t\|$ leads to an inconsistent estimator.

The study on implicit bias for loss functions with the self-bounding property has been very scarce. To our knowledge, [35] crafted the complete hinge loss, which behaves like the hinge loss before GD converges to the zero risk yet incurs an extra penalty to artificially align the parameter toward the max-margin direction. Together with the benefits and caveats of the max-margin implicit bias, we believe this is an interesting open topic.

**Dependency on $n$.** Our main result (Theorem 5) provides the rate depending on the factor $n$. This extra factor with respect to $n$ arises due to the worst-case analysis such that we have *at least* one "bad" direction $\mathbf{z}_i$ before the convergence, corresponding to the inequality (9) in the proof of Theorem 5 (see Section B). This worst-case scenario supposes that all data points are nearly equidistant, which is unlikely since most data points tend to cluster in similar directions. We conjecture that this $n$-dependency is not essential with additional mild data assumptions.

**The stochastic case.** Our result can be extended for the stochastic gradient descent (SGD). Consider the scenario where we sample one fresh data point at each $t$ and update the linear parameter with the loss function computed on this sample. Under the similar setup to Theorem 5, the population risk is $\varepsilon$-optimal with high probability after $T = \Omega(\varepsilon^{-(\alpha+2)})$. The formal statement and proof are shown in Section C. While this rate apparently looks significantly slower than the GD rate $T = \Omega(\varepsilon^{-\alpha})$, the extra iterations $\varepsilon^{-2}$ is necessary for collecting sufficient samples to estimate the population risk. By noting that the GD/SGD updates consume $n$/one samples, the GD/SGD rates are comparable in terms of the number of consumed samples.

**Finite-time convergence.** Last but not least, we may potentially have another benefit of loss functions with separation margin. Take a look at Figure 1 again. Loss functions without separation margin, such as the Tsallis 1.5- and 2-losses, converge to exact zero within finite time when sufficiently large stepsize is used. Such finite-time convergence under the linearly separable case can be shown without significant challenges if we use perceptron, or even the hinge loss, while becoming highly non-trivial in the case of twice-differentiable loss functions. This is because the perceptron argument requires a non-degenerate lower bound on $\langle \mathbf{w}_t, \mathbf{w}_* \rangle$ (see (4)), which is not straightforward therein as the loss gradient can be arbitrarily small positive (due to the twice differentiability of the loss). We conjecture that an additional data assumption is necessary because the loss gradient could be adversarially vanishing against GD convergence, and leave this as future work.

## Acknowledgments

HB is supported by JST PRESTO (Grant No. JPMJPR24K6). SS was supported by JST ERATO (Grant No. JPMJER1903). YT is supported by JSPS KAKENHI (Grant No. 23KJ1336).

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

# A Technical lemmas

We introduce the gradient potential for a loss function in consideration as follows:

$$G(\mathbf{w}) := \frac{1}{n} \sum_{i=1}^{n} g(\langle \mathbf{w}, \mathbf{z}_i \rangle). \tag{6}$$

**Lemma 7.** *Consider a loss $\ell$ satisfying Assumption 3B. Then, we have*

$$\ell\left(\sqrt{\rho(\lambda)}\right) \leq \frac{\rho(\lambda)}{\lambda}.$$

*Proof.* See [63, Lemma 20]. $\qquad\square$

**Lemma 8.** *Consider a Fenchel–Young loss $\ell(z) = \phi^*(-z)$ satisfying Assumption 3A. Then, $\ell$ and $g$ are nonincreasing. Moreover, $\ell$ is strictly decreasing on $(-\infty, m)(\subseteq \mathbb{R})$ if $\ell$ has separation margin $m > 0$; otherwise, $\ell$ is strictly decreasing on $\mathbb{R}$.*

*Proof.* We have by Danskin's theorem [21] $(\phi^*)' = (\phi')^{-1}$, and then

$$\ell'(z) = -(\phi^*)'(-z) = - \underbrace{(\phi')^{-1}(-z)}_{\in \mathrm{dom}(\phi) \subseteq [0,1]} \leq 0,$$

which implies that $\ell$ is nonincreasing. Since $\ell$ is convex and nonincreasing we have that $g(\cdot) = |\ell'(\cdot)| = -\ell'(\cdot)$ is nonincreasing.

For the latter part, if nonincreasing $\ell$ has separation margin, $\ell(z) = 0$ if and only if $z \geq m$. Then, we have $\ell \equiv 0$ on the interval $[m, \infty) \subseteq \mathbb{R}$ and $\ell > 0$ on the interval $(-\infty, m) \subseteq \mathbb{R}$. On the latter interval, $\ell$ must be strictly decreasing because of its convexity. We can prove similarly for $\ell$ lacking separation margin. $\qquad\square$

**Lemma 9.** *Consider a loss $\ell$ satisfying Assumption 3B. Suppose that $\ell$ has separation margin $m > 0$. Then, we have $\rho(\lambda) \leq m^2$ for any $\lambda \geq 1$.*

*Proof.* When $\ell$ has separation margin $m$ (see Definition 3), we have

$$\lambda \ell(z) + z^2 = z^2 \quad \text{for } z \geq m.$$

By the definition of $\rho$, we have

$$\rho(\lambda) = \min_{z \in \mathbb{R}} \lambda \ell(z) + z^2 \leq \min_{z \geq m} z^2 = m^2.$$

$\qquad\square$

**Lemma 10** (Split optimization [63]). *Suppose Assumption 2 and consider a convex and nonincreasing loss $\ell$ satisfying Assumption 3C and let $\mathbf{u} := \mathbf{u}_1 + \mathbf{u}_2$ such that*

$$\mathbf{u}_1 = \theta \mathbf{w}_*, \qquad \mathbf{u}_2 = \frac{\eta C_g}{2\gamma} \mathbf{w}_*.$$

*For every $t \geq 1$, we have*

$$\frac{\|\mathbf{w}_t - \mathbf{u}\|^2}{2\eta t} + \frac{1}{t} \sum_{k=0}^{t-1} L(\mathbf{w}_k) \leq \ell(\gamma\theta) + \frac{1}{2\eta t} \left( \theta + \frac{\eta C_g}{2\gamma} \right)^2.$$

*Proof.* For $k < t$, we have

$$\|\mathbf{w}_{k-1} - \mathbf{u}\|^2 = \|\mathbf{w}_k - \mathbf{u}\|^2 + 2\eta \langle \nabla L(\mathbf{w}_k), \mathbf{u} - \mathbf{w}_k \rangle + \eta^2 \|\nabla L(\mathbf{w}_k)\|^2$$
$$= \|\mathbf{w}_k - \mathbf{u}\|^2 + 2\eta \langle \nabla L(\mathbf{w}_k), \mathbf{u}_1 - \mathbf{w}_k \rangle + \eta(2 \langle \nabla L(\mathbf{w}_k), \mathbf{u}_2 \rangle + \eta \|\nabla L(\mathbf{w}_k)\|^2).$$

For the second term, we have

$$\langle \nabla L(\mathbf{w}_k), \mathbf{u}_1 - \mathbf{w}_k \rangle = \frac{1}{n} \sum_{i=1}^{n} \ell'(\langle \mathbf{w}_k, \mathbf{z}_i \rangle) \langle \mathbf{z}_i, \mathbf{u}_1 - \mathbf{w}_k \rangle$$

$$= \frac{1}{n} \sum_{i=1}^{n} \ell'(\langle \mathbf{w}_k, \mathbf{z}_i \rangle)(\langle \mathbf{u}_1, \mathbf{z}_i \rangle - \langle \mathbf{w}_k, \mathbf{z}_i \rangle) \tag{7}$$

$$\leq \frac{1}{n} \sum_{i=1}^{n} [\ell(\langle \mathbf{u}_1, \mathbf{z}_i \rangle) - \ell(\langle \mathbf{w}_k, \mathbf{z}_i \rangle)] \qquad (\ell: \text{convex})$$

$$\leq \ell(\gamma\theta) - L(\mathbf{w}_k). \qquad (\ell: \text{nonincreasing})$$

For the third term, we have

$$2 \langle \nabla L(\mathbf{w}_k), \mathbf{u}_2 \rangle + \eta \|\nabla L(\mathbf{w}_k)\|^2$$

$$= \frac{2}{n} \sum_{i=1}^{n} \ell'(\langle \mathbf{w}_k, \mathbf{z}_i \rangle) \langle \mathbf{z}_i, \mathbf{u}_2 \rangle + \eta \left\| \frac{1}{n} \sum_{i=1}^{n} \ell'(\langle \mathbf{w}_k, \mathbf{z}_i \rangle)\mathbf{z}_i \right\|^2$$

$$\leq \frac{2}{n} \sum_{i=1}^{n} \ell'(\langle \mathbf{w}_k, \mathbf{z}_i \rangle) \langle \mathbf{z}_i, \mathbf{u}_2 \rangle + \eta \left( \frac{1}{n} \sum_{i=1}^{n} \ell'(\langle \mathbf{w}_k, \mathbf{z}_i \rangle) \right)^2 \qquad (\|\mathbf{z}_i\| \leq 1)$$

$$\leq \frac{2\|\mathbf{u}_2\|}{n} \sum_{i=1}^{n} \ell'(\langle \mathbf{w}_k, \mathbf{z}_i \rangle) \langle \mathbf{z}_i, \mathbf{w}_* \rangle + \eta C_g \cdot G(\mathbf{w}_k) \qquad (\text{Assumption 3C and } G(\mathbf{w}_k) \geq 0)$$

$$\leq -2\gamma \|\mathbf{u}_2\| G(\mathbf{w}_k) + \eta C_g \cdot G(\mathbf{w}_k) \qquad (\text{Assumption 2 and } G(\mathbf{w}_k) \geq 0)$$

$$= 0,$$

where the last equality is by the choice of $\mathbf{u}_2$ and $G(\mathbf{w})$ is defined in (6).

By combining them altogether, we have for $k < t$,

$$\|\mathbf{w}_{k+1} - \mathbf{u}\|^2 \leq \|\mathbf{w}_k - \mathbf{u}\|^2 + 2\eta [\ell(\gamma\theta) - L(\mathbf{w}_k)].$$

Telescoping the sum from 0 to $t - 1$ and rearranging, we get

$$\frac{\|\mathbf{w}_t - \mathbf{u}\|^2}{2\eta t} + \frac{1}{t} \sum_{k=0}^{t-1} L(\mathbf{w}_k) \leq \ell(\gamma\theta) + \frac{\|\mathbf{w}_0 - \mathbf{u}\|^2}{2\eta t},$$

which completes the proof. $\qquad\square$

## B   Proof of Theorem 5

**Lemma 11.** *Suppose Assumption 2 and consider* (GD) *with any stepsize $\eta > 0$ under a Fenchel–Young loss $\ell$ that satisfies Assumption 3A. For $t \geq 1$, assume $G(\mathbf{w}_k) \geq G_{\min} > 0$ for all $k \in [0, t-1]$, where $G(\mathbf{w})$ is defined in (6). Then, we have*

$$\gamma\eta G_{\min} t \leq \langle \mathbf{w}_t, \mathbf{w}_* \rangle - \langle \mathbf{w}_0, \mathbf{w}_* \rangle.$$

*Proof.* By the perceptron argument [44], we have

$$\langle \mathbf{w}_{k+1}, \mathbf{w}_* \rangle = \langle \mathbf{w}_k, \mathbf{w}_* \rangle - \eta \langle \nabla L(\mathbf{w}_k), \mathbf{w}_* \rangle$$

$$= \langle \mathbf{w}_k, \mathbf{w}_* \rangle - \frac{\eta}{n} \left\langle \sum_{i=1}^{n} \ell'(\langle \mathbf{w}_k, \mathbf{z}_i \rangle)\mathbf{z}_i, \mathbf{w}_* \right\rangle$$

$$= \langle \mathbf{w}_k, \mathbf{w}_* \rangle + \frac{\eta}{n} \sum_{i=1}^{n} g(\langle \mathbf{w}_k, \mathbf{z}_i \rangle) \langle \mathbf{w}_*, \mathbf{z}_i \rangle \qquad (\text{use } g(\cdot) = -\ell'(\cdot) \text{ by Lemma 8})$$

$$\geq \langle \mathbf{w}_k, \mathbf{w}_* \rangle + \gamma\eta G(\mathbf{w}_k) \qquad (\text{note } g(\cdot) \geq 0 \text{ and Assumption 2})$$

$$\geq \langle \mathbf{w}_k, \mathbf{w}_* \rangle + \gamma\eta G_{\min}.$$

Telescoping the sum, we have the desired inequality. $\qquad\square$

**Lemma 12** (Order evaluation). *Let $\mathcal{I} \subseteq \mathbb{R}_{\geq 0}$ be an open interval containing zero as the left end. For $f \colon \mathbb{R}_{\geq 0} \to \mathbb{R}_{\geq 0} \cup \{\infty\}$ that is nondecreasing and differentiable on $\mathcal{I}$ and satisfies $f(0) = 0$, let*

$$\alpha := \sup_{x \in (0, x_0]} \frac{x f'(x)}{f(x)} \quad \text{for some } x_0 \in \mathcal{I}. \tag{8}$$

*Then, for any $x \in (0, x_0)$, we have*

$$f(x) \geq C x^\alpha, \quad \text{where} \quad C := \frac{f(x_0)}{x_0^\alpha}.$$

*Proof.* By the definition of $\alpha$, we have

$$\alpha \geq \frac{x f'(x)}{f(x)} \quad \text{for all } x \in (0, x_0].$$

Then, for any $x \in (0, x_0)$, we have

$$\alpha \ln \frac{x_0}{x} = \alpha \int_x^{x_0} \frac{\mathrm{d}s}{s} \geq \int_x^{x_0} \frac{f'(s)}{f(s)} \mathrm{d}s = \ln \frac{f(x_0)}{f(x)}.$$

By reorganizing this inequality, we can prove the original argument. $\square$

**Lemma 13.** *Consider a Fenchel–Young loss $\ell(z) = \phi^*(-z)$ satisfying Assumption 3A. Then, for arbitrary $0 < \bar{\varepsilon} < 1$ and $\alpha$ defined in (3), we have*

$$g(\ell^{-1}(\varepsilon)) \geq C_\phi \varepsilon^\alpha \quad \text{for } \varepsilon \in (0, \bar{\varepsilon}), \quad \text{where} \quad C_\phi := \frac{g(\ell^{-1}(\bar{\varepsilon}))}{\bar{\varepsilon}^\alpha}.$$

*Proof.* Choose any $\varepsilon_0 > 0$. For $\varepsilon \in (0, \varepsilon_0)$, we can invert to get $z \equiv \ell^{-1}(\varepsilon)$ because $\ell$ is strictly decreasing on $\ell^{-1}((0, 1))$ (by Lemma 8) and hence invertible. Note that

$$\begin{cases} z \in (\ell^{-1}(\varepsilon_0), m) & \text{if } \ell \text{ has separation margin } m > 0, \\ z \in (\ell^{-1}(\varepsilon_0), \infty) & \text{otherwise,} \end{cases}$$

because $\ell^{-1}(\cdot)$ is nonincreasing. We write this range as $\mathcal{I}$, then $z = \ell^{-1}(\varepsilon) \in \mathcal{I}$ for $\varepsilon \in (0, \varepsilon_0)$.

Let us verify $g(z) = -\ell'(z)$ is differentiable at $z \in \mathcal{I}$ first. By the definition of Fenchel–Young losses, we have

$$\ell(z) = \phi^*(-z) = \sup_{x \in (0,1)} [x \cdot (-z) - \phi(x)] = -z \cdot (\phi')^{-1}(-z) - \phi\left((\phi')^{-1}(-z)\right),$$

for $z \in \mathcal{I}$, where we used the first-order optimality $-z = \phi'(x)$ of the convex conjugate at the last identity. Since $\phi$ is twice continuously differentiable and $\phi'' > 0$ on the interval $(0, 1)$ by Assumption 3A, we can apply the inverse function theorem to have

$$\ell'(z) = -(\phi')^{-1}(-z) - \frac{z}{\phi''\left((\phi')^{-1}(-z)\right)} - \frac{\phi'\left((\phi')^{-1}(-z)\right)}{\phi''\left((\phi')^{-1}(-z)\right)} = -(\phi')^{-1}(-z),$$

for $z \in \mathcal{I}$. Since $\phi$ is twice continuously differentiable, we can apply the inverse function theorem once again to get $\ell''(z)$ for $z \in \mathcal{I}$, and hence $g$ is differentiable on $\mathcal{I}$.

In addition, $\ell$ is continuously differentiable with non-degenerate derivative at $z \in \mathcal{I}$ because $\ell$ is strictly decreasing on $\mathcal{I}$. From these observations, we can see that $g(\ell^{-1}(\cdot)) =: f(\cdot)$ is nondecreasing on $\mathcal{I}$ and differentiable on $\mathcal{I}$ (because it is the composition of two nondecreasing and differentiable functions $g$ and $\ell^{-1}$). Now we can apply Lemma 12 to this $f$. Let us compute the exponent $\alpha_\varepsilon$ defined in (8). By the differentiability and non-degenerate derivative of $\ell$ on $\mathcal{I}$, we can apply the inverse function theorem on $\ell$ to have

$$\frac{\varepsilon f'(\varepsilon)}{f(\varepsilon)} = \frac{\varepsilon}{g(\ell^{-1}(\varepsilon))} \cdot g'(\ell^{-1}(\varepsilon)) \cdot \frac{1}{\ell'(\ell^{-1}(\varepsilon))} \qquad \text{(inverse function theorem)}$$

$$= \frac{\ell(z) g'(z)}{g(z) \ell'(z)} \qquad (\varepsilon \equiv \ell(z))$$

$$= \frac{\ell(z)\ell''(z)}{[\ell'(z)]^2} \qquad\qquad (g(\cdot) = -\ell'(\cdot))$$

$$= \frac{\phi^*(\bar{z}) \cdot (\phi^*)''(\bar{z})}{[(\phi^*)'(\bar{z})]^2} \qquad\qquad (\bar{z} := -z)$$

$$\overset{(A)}{=} \frac{[\mu\phi'(\mu) - \phi(\mu)] \cdot \frac{1}{\phi''(\mu)}}{\mu^2} \qquad\qquad (\mu \equiv (\phi^*)'(\bar{z}))$$

$$= \frac{\phi'(\mu)}{\mu\phi''(\mu)} \left[ 1 - \frac{\phi(\mu)}{\mu\phi'(\mu)} \right],$$

where at (A) we introduce $\mu$ as the dual of $\bar{z}$ by the mirror map $\phi'$ such that

$$\bar{z} = \phi'(\mu) \quad \text{and} \quad \mu = (\phi^*)'(\bar{z}),$$

which implies $\phi^*(\bar{z}) = \mu\phi'(\mu) - \phi(\mu)$ together with the definition of the convex conjugate, and

$$[\phi''(\mu)] \cdot [(\phi^*)''(\bar{z})] = 1$$

with Danskin's theorem [21] and the inverse function theorem. Note that this identity is often referred to as Crouzeix's identity [19]. Here, we have

$$\begin{cases} \bar{z} \in \left(-m, -\ell^{-1}(\varepsilon_0)\right) & \text{and } \mu \in \left(0, g(\ell^{-1}(\varepsilon_0))\right) & \text{if } \ell \text{ has separation margin } m > 0, \\ \bar{z} \in \left(-\infty, -\ell^{-1}(\varepsilon_0)\right) & \text{and } \mu \in \left(0, g(\ell^{-1}(\varepsilon_0))\right) & \text{otherwise,} \end{cases}$$

by noting that $g(z) = -\ell'(z) = (\phi^*)'(-z)$ is nonincreasing. With this primal-dual relationship, we have

$$\phi^*(\bar{z}) = \mu\bar{z} - \phi(\mu) \quad \text{and} \quad [\phi''(\mu)] \cdot [(\phi^*)''(\bar{z})] = 1$$

by the definition of the convex conjugate and Crouzeix's identity. Now we are ready to apply Lemma 12, which yields

$$g(\ell^{-1}(\varepsilon)) \geq C_\phi \varepsilon^\alpha \quad \text{for } \varepsilon \in (0, \bar{\varepsilon}),$$

where

$$\alpha := \sup_{\mu \in (0, \bar{s}]} \frac{\phi'(\mu)}{\mu\phi''(\mu)} \left[ 1 - \frac{\phi(\mu)}{\mu\phi'(\mu)} \right], \quad C_\phi := \frac{g(\ell^{-1}(\bar{\varepsilon}))}{\bar{\varepsilon}^\alpha}, \quad \text{and} \quad \bar{\varepsilon} := g(\ell^{-1}(\varepsilon_0)).$$

Since the choice of $\varepsilon_0 > 0$ was arbitrary and $\text{Im}(g) = \text{Im}((\phi^*)') = \text{dom}(\phi') \subseteq [0,1]$, we can choose such $\bar{\varepsilon} \in (0,1)$. $\qquad\qquad\square$

*Proof of Theorem 5.* For a fixed $k \in [T-1]$, if we have $L(\mathbf{w}_k) > \varepsilon$, there exists $i \in [n]$ such that $\ell(\langle \mathbf{w}_k, \mathbf{z}_i \rangle) > \varepsilon$. Then, we have for this specific $i \in [n]$,

$$\langle \mathbf{w}_k, \mathbf{z}_i \rangle < \ell^{-1}(\varepsilon) \qquad (\ell \text{ is strictly decreasing when } \ell > 0 \text{ by Lemma 8})$$
$$\implies g(\langle \mathbf{w}_k, \mathbf{z}_i \rangle) \geq g(\ell^{-1}(\varepsilon)). \quad (g \text{ is nonincreasing by Lemma 8})$$

This implies that

$$G(\mathbf{w}_k) = \frac{1}{n} \sum_{j \in [n]} g(\langle \mathbf{w}_k, \mathbf{z}_j \rangle) \geq \frac{1}{n} g(\langle \mathbf{w}_k, \mathbf{z}_i \rangle) \geq \frac{1}{n} g\left(\ell^{-1}(\varepsilon)\right) \qquad (9)$$

holds while $L(\mathbf{w}_k)$ is $\varepsilon$-suboptimal, that is, $L(\mathbf{w}_k) > \varepsilon$.

Next, fix $\mathbf{w}_0 = \mathbf{0}$ and consider the case where $L(\mathbf{w}_k) > \varepsilon$ holds for all $k \in [T-1]$. By Lemma 8, we can use Lemma 19. By Lemmas 11 and 19, we can take $G_{\min} = g(\ell^{-1}(\varepsilon))/n$ (noting (9)) and have

$$\frac{\gamma\eta g\left(\ell^{-1}(\varepsilon)\right) T}{n} \leq \langle \mathbf{w}_T, \mathbf{w}_* \rangle - \langle \mathbf{w}_0, \mathbf{w}_* \rangle \quad \text{(Lemma 11)}$$

$$= \langle \mathbf{w}_T, \mathbf{w}_* \rangle \qquad\qquad \text{(with the choice of } \mathbf{w}_0 = \mathbf{0}\text{)}$$

$$\leq \|\mathbf{w}_T\| \qquad\qquad \text{(the Cauchy–Schwarz inequality with } \|\mathbf{w}_*\| = 1\text{)}$$

$$\leq \frac{4\sqrt{\rho(\gamma^2\eta T)} + \eta C_g}{\gamma}. \quad \text{(Lemma 19)}$$

By reorganizing and applying Lemma 13, we leverage the primal-dual relationship to have

$$T \leq \frac{n}{\gamma^2} \left( \frac{4\sqrt{\rho(\gamma^2 \eta T)}}{\eta} + C_g \right) \cdot \frac{1}{g\left(\ell^{-1}(\varepsilon)\right)} \leq \frac{n}{\gamma^2} \left( \frac{4\sqrt{\rho(\gamma^2 \eta T)}}{\eta} + C_g \right) \cdot \frac{1}{C_\phi \varepsilon^\alpha},$$

where $C_\phi$ is defined in Lemma 13. Therefore, $L(\mathbf{w}_k)$ is $\varepsilon$-suboptimal after at most

$$\frac{n}{C_\phi \gamma^2} \left( \frac{4\sqrt{\rho(\gamma^2 \eta t)}}{\eta} + C_g \right) \varepsilon^{-\alpha} \qquad (=: T(\varepsilon))$$

iterations. That is, if $T > T(\varepsilon)$, the gradient lower bound (9) must be violated at some $t \in [T]$, and for this $t$, we achieve $L(\mathbf{w}_t) \leq \varepsilon$.

Finally, we verify that $C_\phi$ defined in Lemma 13 matches (3). By introducing $\bar{z}$ as the dual of $\bar{\mu}$ such that

$$\bar{z} = \phi'(\bar{\mu}) \quad \text{and} \quad \bar{\mu} = (\phi^*)'(\bar{z}),$$

we have

$$\begin{aligned}
\bar{\varepsilon} &= \ell(g^{-1}(\bar{\mu})) && (g \text{ is invertible when } 0 < g(\cdot) < 1) \\
&= \phi^*(\bar{z}) && (\bar{\mu} = (\phi^*)'(\bar{z}) = g(-\bar{z}) \text{ implies } g^{-1}(\bar{\mu}) = -\bar{z}) \\
&= \bar{\mu}\phi'(\bar{\mu}) - \phi(\bar{\mu}),
\end{aligned}$$

where the invertibility of $g$ can be verified through the differentiability of $g$ as in Lemma 13 (by relying upon $\phi'' > 0$ in Assumption 3A), and the last identity follows by the definition of the convex conjugate. Plugging this into $C_\phi$ defined in Lemma 13, we see that it matches $C_\phi$ in (3). Thus, we have proven all statements. □

## C   Extension to the stochastic gradient descent

We discuss the extension of Theorem 5 to the stochastic setup. We consider the constant-stepsize online stochastic gradient descent (SGD) as follows:

$$\mathbf{w}_{t+1} := \mathbf{w}_t - \eta \nabla L_t(\mathbf{w}_t), \quad \text{where} \quad L_t(\mathbf{w}) := \ell(\langle \mathbf{w}, y_t \mathbf{x}_t \rangle), \quad t \geq 0, \qquad \text{(SGD)}$$

for a loss function $\ell \colon \mathbb{R} \to \mathbb{R}_{\geq 0}$. Here, $(\mathbf{x}_t, y_t)_{t \geq 0}$ are independently and identically distributed according to the following assumption.

**Assumption 4.** *Assume the training data $(\mathbf{x}_t, y_t)_{t \geq 0}$ are independent copies of $(\mathbf{x}, y)$ following a distribution such that*

1. *$\|\mathbf{x}\| \leq 1$, and $y_i \in \{\pm 1\}$, almost surely;*

2. *there is $\gamma > 0$ and a unit vector $\mathbf{w}_*$ such that $\langle \mathbf{w}_*, \mathbf{z}_t \rangle \geq \gamma$ for $\mathbf{z}_t := y_t \mathbf{x}_t$, almost surely.*

**Proposition 14.** *Suppose Assumption 4 and consider* (SGD) *with stepsize $\eta > 0$ and $\mathbf{w}_0 = \mathbf{0}$ under a Fenchel–Young loss $\ell$ satisfying Assumption 3, and additionally having separation margin $m > 0$. For arbitrary $\delta, \bar{\varepsilon} \in (0, 1)$, define $(\alpha, C_\phi)$ as in Eq. (3), for which we assume $\alpha, C_\phi \in (0, \infty)$. In addition, for arbitrary $\varepsilon \in (0, \bar{\varepsilon})$, define*

$$M_{\eta, \gamma} := \ell\left( -\frac{4m + \eta C_g}{\gamma} \right) \quad \text{and} \quad t^\circ := \max\left\{ \frac{32 M_{\eta, \gamma}^2 \ln(1/\delta)}{\varepsilon^2}, \frac{8 M_{\eta, \gamma} \ln(1/\delta)}{\varepsilon} \right\}.$$

*Then, if we run* (SGD) *with $T$ iterations such that*

$$T > N \cdot t^\circ, \quad \text{where} \quad N := \frac{2^\alpha}{C_\phi \gamma^2} \left( \frac{4m}{\eta} + C_g \right) \varepsilon^{-\alpha},$$

*then we have $\min_{t \in [T]} \mathbb{E}[L_t(\mathbf{w}_t)] \leq \varepsilon$ with probability at least $(1 - \delta)^N$.*

Before proving Proposition 14, several auxiliary lemmas are presented. Since the proof of Proposition 14 closely follows Theorem 5, the following Lemmas 15 and 16 are almost identical to the deterministic versions (Lemmas 11 and 19, respectively), and hence we omit the proofs.

**Lemma 15.** *Suppose Assumption 4 and consider* (SGD) *with any stepsize $\eta > 0$ under a Fenchel–Young loss that satisfies Assumption 3A. Moreover, assume that there exists $k \in [0, t-1]$ such that we have a non-trivial lower bound $g(\langle \mathbf{w}_k, \mathbf{z}_k \rangle) \geq g_{\min} > 0$. Then, we have*

$$\gamma \eta g_{\min} \leq \langle \mathbf{w}_t, \mathbf{w}_* \rangle - \langle \mathbf{w}_0, \mathbf{w}_* \rangle.$$

**Lemma 16.** *Suppose Assumption 4 and consider* (SGD) *with any stepsize $\eta > 0$ under a convex and nondecreasing loss $\ell$ satisfying Assumptions 3B and 3C. For every $t \geq 1$, we have*

$$\|\mathbf{w}_t\| \leq \frac{4\sqrt{\rho(\gamma^2 \eta t)} + \eta C_g}{\gamma}.$$

The following concentration result is an additional argument that we need in the stochastic case.

**Lemma 17.** *Consider* (SGD) *with any stepsize $\eta > 0$ under a Fenchel–Young loss that satisfies Assumptions 3A and 3C. Let $Z_0, \ldots, Z_{t-1}$ be the independent copies of data $\mathbf{z} = y\mathbf{x}$ following Assumption 4. We introduce the filtration $\{\mathcal{F}_t\}_{t \geq 0}$, where $\mathcal{F}_k$ is a $\sigma$-algebra defined on $Z_1, \ldots, Z_k$, and let $H_k := \ell(\langle \mathbf{w}_k, Z_k \rangle)$ be a random variable, where $\mathbf{w}_k$ is an $\mathcal{F}_{k-1}$-measurable random variable. We write $S_t := \sum_{k=0}^{t-1} H_k$ for a random variable standing for the accumulated loss, and let $\overline{S}_t := S_t/t$. Moreover, we introduce the following assumptions:*

- *(**Bounded mean**) $\mu_k := \mathbb{E}[H_k | \mathcal{F}_{k-1}] \leq M$ for $k = 0, \ldots, t-1$.*
- *(**Bounded variance**) $\mathrm{Var}(H_k | \mathcal{F}_{k-1}) \leq \sigma^2$ for $k = 0, \ldots, t-1$.*

*Then, for any $\varepsilon > 0$ and $\delta \in (0, 1)$, if we have*

$$\frac{\mathbb{E}[S_t]}{t} > \varepsilon \quad and \quad t \geq \max\left\{ \frac{32\sigma^2 \ln(1/\delta)}{\varepsilon^2}, \frac{8M \ln(1/\delta)}{\varepsilon} \right\}, \tag{10}$$

*then we have $\overline{S}_t > \varepsilon/2$ with probability at least $1 - \delta$.*

*Proof.* First, we apply a type of the martingale inequality, *Freedman's inequality* [22, Theorem 1.6]. Since $\mathbb{E}[H_k - \mu_k | \mathcal{F}_{k-1}]$ is the mean-zero martingale difference with the bounded variance $\sigma^2$, Freedman's inequality is applicable. Then, we have

$$\Pr\{ S_t \leq \mathbb{E}[S_t] - u \} \leq \exp\left[ -\frac{u^2}{2(Mu + t\sigma^2)} \right].$$

Equivalently, we have the following inequality with probability at least $1 - \delta$:

$$S_t > \mathbb{E}[S_t] - \sqrt{M^2 \ln^2(1/\delta) + 2t\sigma^2 \ln(1/\delta)} - M \ln(1/\delta)$$
$$> \mathbb{E}[S_t] - \sqrt{2t\sigma^2 \ln(1/\delta)} - 2M \ln(1/\delta).$$

Dividing by $t$, we have

$$\overline{S}_t > \frac{\mathbb{E}[S_t]}{t} - \sqrt{\frac{2\sigma^2 \ln(1/\delta)}{t}} - \frac{2M \ln(1/\delta)}{t} > \varepsilon - \frac{\varepsilon}{4} - \frac{\varepsilon}{4} = \frac{\varepsilon}{2},$$

where we used the conditions (10) at the second inequality. Thus, the desired inequality is shown. $\square$

Now we are ready to prove Proposition 14. Overall, the proof consists of the perceptron argument and the concentration property.

*Proof of Proposition 14.* The first step is to establish the concentration property. To apply Lemma 17, we confirm the bounded moment conditions. For the mean $\mathbb{E}[H_k | \mathcal{F}_{k-1}] = \mathbb{E}[\ell(\langle \mathbf{w}_k, Z_k \rangle) | \mathcal{F}_{k-1}]$, where $\mathcal{F}_{k-1}$ is the $\sigma$-algebra defined on $\{Z_l\}_{l=1}^{k-1}$, we have

$$\mathbb{E}[\ell(\langle \mathbf{w}_k, Z_k \rangle) | \mathcal{F}_{k-1}] \leq \mathbb{E}[\ell(-\|\mathbf{w}_k\|) | \mathcal{F}_{k-1}] \quad (\ell \text{ is nonincreasing and } \|Z_k\| \leq 1)$$
$$\leq \ell\left( -\frac{4m + \eta C_g}{\gamma} \right) \quad \text{(Lemmas 9 and 15)}$$
$$=: M_{\eta, \gamma}.$$

For the variance $\mathrm{Var}(H_k|\mathcal{F}_{k-1})$, we similarly have

$$\begin{aligned}
\mathrm{Var}(H_k|\mathcal{F}_{k-1}) &= \mathbb{E}[H_k^2|\mathcal{F}_{k-1}] - \mathbb{E}[H_k|\mathcal{F}_{k-1}]^2 \\
&\leq \mathbb{E}[H_k^2|\mathcal{F}_{k-1}] \\
&\leq \ell\left(-\frac{4m + \eta C_g}{\gamma}\right)^2 \\
&= M_{\eta,\gamma}^2.
\end{aligned}$$

Note that these moment bounds hold uniformly for any $k$. By plugging $M = M_{\eta,\gamma}$ and $\sigma^2 = M_{\eta,\gamma}^2$ into Lemma 17, if we have

$$\frac{1}{t^\circ}\sum_{k=t_0}^{t_0+t^\circ-1}\mathbb{E}\left[L_k(\mathbf{w}_k)\right] > \varepsilon \quad \text{and} \quad t^\circ \geq \max\left\{\frac{32M_{\eta,\gamma}^2\ln(1/\delta)}{\varepsilon^2}, \frac{8M_{\eta,\gamma}\ln(1/\delta)}{\varepsilon}\right\}, \quad (11)$$

then we have

$$\frac{1}{t^\circ}\sum_{k=t_0}^{t_0+t^\circ-1}L_k(\mathbf{w}_k) > \frac{\varepsilon}{2} \quad \text{with probability at least } 1-\delta.$$

Let us use this concentration argument. Split the interval $[0,T]$ into length-$t^\circ$ sub-intervals (for $t^\circ$ satisfying (11)) such that

$$[0,T] = \underbrace{[0,t^\circ-1]}_{=:\mathcal{I}_1}\sqcup\underbrace{[t^\circ,2t^\circ-1]}_{=:\mathcal{I}_2}\sqcup\underbrace{[2t^\circ,3t^\circ-1]}_{=:\mathcal{I}_3}\cdots\sqcup\underbrace{[(N-1)t^\circ,Nt^\circ-1]}_{=:\mathcal{I}_N}\sqcup[Nt^\circ,T]. \quad (12)$$

Consider the scenario where $\mathbb{E}[L_k(\mathbf{w}_k)] > \varepsilon$ holds for all $k \in [0,T]$, and focus on an arbitrary sub-interval $\mathcal{I}_l$. Since we have $\frac{1}{t^\circ}\sum_{k\in\mathcal{I}_l}\mathbb{E}[L_k(\mathbf{w}_k)] > \varepsilon$, the concentration argument implies that $\frac{1}{t^\circ}\sum_{k\in\mathcal{I}_l}L_k(\mathbf{w}_k) > \varepsilon/2$ with probability at least $1-\delta$. This further indicates the high-probability existence of $k_l \in \mathcal{I}_l$ such that $L_{k_l}(\mathbf{w}_{k_l}) > \varepsilon/2$. In this case, we have $g(\langle\mathbf{w}_{k_l},\mathbf{z}_{k_l}\rangle) \geq g(\ell^{-1}(\varepsilon/2))$ for this specific $k_l \in \mathcal{I}_l$, which can be seen in the same manner as the proof of Theorem 5. Since this concentration argument does not depend on the sub-interval choice $\mathcal{I}_l$, there exists a set of indices $\{k_l\}_{l\in[N]}$ such that each of $g(\langle\mathbf{w}_{k_l},\mathbf{z}_{k_l}\rangle) \geq g(\ell^{-1}(\varepsilon/2))$ holds with probability at least $1-\delta$.

Next, we invoke the perceptron argument. By combining Lemmas 15 and 16 with the choice $g_{\min} = g(\ell^{-1}(\varepsilon/2))$, we have

$$\gamma\eta g\left(\ell^{-1}\left(\frac{\varepsilon}{2}\right)\right)\cdot N \leq \langle\mathbf{w}_T,\mathbf{w}_*\rangle \leq \|\mathbf{w}_T\| \leq \frac{4\sqrt{\rho(\gamma^2\eta T)} + \eta C_g}{\gamma} \leq \frac{4m + \eta C_g}{\gamma},$$

with probability at least $(1-\delta)^N$, where we additionally used Lemma 9 at the last inequality. By applying Lemma 13 at the left-most side, we have

$$\gamma\eta C_\phi \cdot \left(\frac{\varepsilon}{2}\right)^\alpha \cdot N \leq \frac{4m + \eta C_g}{\gamma},$$

which implies

$$N \leq \frac{2^\alpha}{C_\phi\gamma^2}\left(\frac{4m}{\eta} + C_g\right)\varepsilon^{-\alpha} \quad (=: N(\varepsilon)).$$

Therefore, if $N > N(\varepsilon)$ (or $T > N(\varepsilon)\cdot t^\circ$), with probability at least $(1-\delta)^N$, we have $\mathbb{E}[L_k(\mathbf{w}_k)] \leq \varepsilon$ for some $k \in [0,T]$.

Finally, we need verify that $C_\phi$ defined in Lemma 13 matches (3), but we skip it because it can be confirmed in the same manner as in the proof of Theorem 5. $\square$

## C.1 Comparison between GD and SGD

Whereas the iteration complexity for GD given by Corollary 6 is $T = \Omega(\varepsilon^{-\alpha})$, the iteration complexity for SGD given by Proposition 14 is $T = \Omega(\varepsilon^{-(\alpha+2)})$. Hence, the SGD rate is significantly slower than the GD case. This deterioration is because we can observe a non-trivial lower bound $g(\langle\mathbf{w}_k,\mathbf{z}_k\rangle)$

after every $t^\circ = \Omega(\varepsilon^{-2})$ steps. As in the GD case, we need $N = \Omega(\varepsilon^{-\alpha})$ such non-trivial lower bounds, and hence the total iteration number amounts to $Nt^\circ = \Omega(\varepsilon^{-(\alpha+2)})$. While this apparently looks a big bottleneck, note that the GD rate is $T \gtrsim n\varepsilon^{-\alpha}$ if we explicitly write the $n$-dependency. Since (SGD) consumes only one fresh sample at every update (while (GD) consumes $n$ samples), the extra complexity $N = \Omega(\varepsilon^{-2})$ appearing in the SGD case compensates for this gap of sample sizes. The GD/SGD rates are comparable in this sense.

## D  Phase transition of large-stepsize GD

We recap Wu et al. [63], who shows the existence of the phase transition from the EoS to stable phases.

**Assumption 5.** *Consider a loss $\ell \in \mathcal{C}^1(\mathbb{R})$ that is convex, nonincreasing, and $\ell(+\infty) = 0$.*

A. **Self-bounding property.** *For some $C_\beta > 0$, $g(\cdot) \le C_\beta \ell(\cdot)$ and*

$$\ell(z) \le \ell(x) + \ell'(x)(z-x) + C_\beta g(x)(z-x)^2 \quad \textit{for } z \textit{ and } x \textit{ such that } |z-x| < 1.$$

B. **Exponential tail.** *There is a constant $C_e > 0$ such that $\ell(z) \le C_e g(z)$ for $z \ge 0$.*

**Theorem 18** ([63]). *Consider (GD) with stepsize $\eta > 0$ and initialization $\mathbf{w}_0 = \mathbf{0}$ under a loss $\ell$ satisfying Assumptions 3B and 3C, and 5A. Let $T$ be the maximum number of steps. Then, we have the following:*

- **The EoS phase.** *For every $t > 0$, we have*

$$\frac{1}{t} \sum_{k=0}^{t-1} L(\mathbf{w}_k) \le \frac{[6\sqrt{\rho(\gamma^2 \eta t)} + \eta C_g]^2}{8\gamma^2 \eta t}.$$

- **The stable phase.** *If $s < T$ is such that*

$$L(\mathbf{w}_s) \le \min\left\{ \frac{1}{4C_\beta^2 \eta}, \frac{\ell(0)}{n} \right\}, \tag{13}$$

*then (GD) is in the stable phase, that is, $(L(\mathbf{w}_t))_{t=s}^T$ decreases monotonically, and moreover,*

$$L(\mathbf{w}_t) \le 5\frac{\rho(\gamma^2 \eta(t-s))}{\gamma^2 \eta(t-s)}, \quad t \in (s, T].$$

- **Phase transition time.** *There exists a constant $C_1 > 0$ that only depends on $C_g$, $C_\beta$, and $\ell(0)$ such that the following holds. Let*

$$\tau := \frac{1}{\gamma^2} \max\left\{ \frac{\psi^{-1}(C_1(\eta+n))}{\eta}, C_1(\eta+n)\eta \right\}, \quad \textit{where} \quad \psi(\lambda) := \frac{\lambda}{\rho(\lambda)}.$$

*If $\tau \le T$, (13) holds for some $s \le \tau$. Moreover, if $\ell$ additionally satisfies Assumption 5B and $\eta \ge 1$, there exists $C_2 > 0$ that depends on $C_e$, $C_g$, $C_\beta$, $\ell(0)$, and $n$ such that $\tau$ is improved as follows:*

$$\tau := \frac{C_2}{\gamma^2} \max\left\{\eta, n\right\}.$$

The proof consists of Lemma 19 (the EoS phase), Lemma 23 (the stable phase), and Lemmas 24 and 25 (phase transition time), respectively. Most of the results in this section have already been provided in Wu et al. [63]. We restate the statements and proofs here to make the paper self-contained, and moreover, simplify the statements from the NTK setup to the linear-model case to highlight the essential structures.

**Lemma 19** (EoS phase). *Suppose Assumption 2 and consider a convex and nonincreasing loss $\ell$ satisfying Assumptions 3B and 3C. For every $t \ge 1$, we have*

$$\frac{1}{t} \sum_{k=0}^{t-1} L(\mathbf{w}_k) \le \frac{[6\sqrt{\rho(\gamma^2 \eta t)} + \eta C_g]^2}{8\gamma^2 \eta t},$$

*and*

$$\|\mathbf{w}_t\| \le \frac{4\sqrt{\rho(\gamma^2 \eta t)} + \eta C_g}{\gamma}.$$

*Proof.* By invoking Lemma 10 with the choice of $\theta$

$$\theta = \frac{\sqrt{\rho(\gamma^2 \eta t)}}{\gamma},$$

we have

$$\frac{\|\mathbf{w}_t - \mathbf{u}\|^2}{2\eta t} + \frac{1}{t}\sum_{k=0}^{t-1} L(\mathbf{w}_k) \leq \ell(\gamma\theta) + \frac{1}{2\eta t}\left(\theta + \frac{\eta C_g}{2\gamma}\right)^2$$

$$\leq \frac{\rho(\gamma^2 \eta t)}{\gamma^2 \eta t} + \frac{1}{2\eta t}\left(\theta + \frac{\eta C_g}{2\gamma}\right)^2, \quad \text{(Lemma 7)}$$

which implies that

$$\|\mathbf{w}_t\| \leq \|\mathbf{w}_t - \mathbf{u}\| + \|\mathbf{u}\|$$

$$\leq \sqrt{\frac{2\rho(\gamma^2 \eta t)}{\gamma^2} + \left(\theta + \frac{\eta C_g}{2\gamma}\right)^2} + \left(\theta + \frac{\eta C_g}{2\gamma}\right)$$

$$\leq \frac{\sqrt{2\rho(\gamma^2 \eta t)}}{\gamma} + 2\left(\theta + \frac{\eta C_g}{2\gamma}\right) \qquad (\sqrt{a+b} \leq \sqrt{a} + \sqrt{b})$$

$$\leq \frac{4\sqrt{\rho(\gamma^2 \eta t)} + \eta C_g}{\gamma},$$

and

$$\frac{1}{t}\sum_{k=0}^{t-1} L(\mathbf{w}_k) \leq \frac{\rho(\gamma^2 \eta t)}{\gamma^2 \eta t} + \frac{1}{2\eta t}\left(\theta + \frac{\eta C_g}{2\gamma}\right)^2$$

$$= \frac{\rho(\gamma^2 \eta t)}{\gamma^2 \eta t} + \frac{(2\sqrt{\rho(\gamma^2 \eta t)} + \eta C_g)^2}{8\gamma^2 \eta t}$$

$$\leq \frac{[2(1+\sqrt{2})\sqrt{\rho(\gamma^2 \eta t)} + \eta C_g]^2}{8\gamma^2 \eta t} \qquad (a^2 + b^2 \leq (a+b)^2 \text{ for } a, b \geq 0)$$

$$\leq \frac{[6\sqrt{\rho(\gamma^2 \eta t)} + \eta C_g]^2}{8\gamma^2 \eta t}.$$

Thus, the proof is completed. $\qquad\square$

**Lemma 20.** *Suppose Assumption 2 and consider a convex and nonincreasing loss $\ell$ satisfying Assumptions 3B and 3C. Then, we have*

$$\frac{1}{t}\sum_{k=0}^{t-1} G(\mathbf{w}_k) \leq \frac{4\sqrt{\rho(\gamma^2 \eta t)} + \eta C_g}{\gamma^2 \eta t}, \quad t \leq T,$$

*where $G(\mathbf{w})$ is defined in (6).*

*Proof.* By the perceptron argument [44], we have

$$\langle \mathbf{w}_{t+1}, \mathbf{w}_* \rangle = \langle \mathbf{w}_t, \mathbf{w}_* \rangle - \eta \langle \nabla L(\mathbf{w}_t), \mathbf{w}_* \rangle$$

$$= \langle \mathbf{w}_t, \mathbf{w}_* \rangle - \frac{\eta}{n}\sum_{i=1}^{n} \ell'(\langle \mathbf{w}_t, \mathbf{z}_i \rangle) \langle \mathbf{w}_*, \mathbf{z}_i \rangle$$

$$\geq \langle \mathbf{w}_t, \mathbf{w}_* \rangle - \frac{\gamma\eta}{n}\sum_{i=1}^{n} \ell'(\langle \mathbf{w}_t, \mathbf{z}_i \rangle) \qquad \text{(Assumption 2 and note } -\ell'(\cdot) \geq 0)$$

$$= \langle \mathbf{w}_t, \mathbf{w}_* \rangle - \gamma\eta G(\mathbf{w}_t).$$

Telescoping the sum, we have

$$\frac{1}{t}\sum_{k=0}^{t-1} G(\mathbf{w}_k) \leq \frac{\langle \mathbf{w}_t, \mathbf{w}_* \rangle - \langle \mathbf{w}_0, \mathbf{w}_* \rangle}{\gamma\eta t} \leq \frac{\|\mathbf{w}_t\|}{\gamma\eta t} \leq \frac{4\sqrt{\rho(\gamma^2 \eta t)} + \eta C_g}{\gamma^2 \eta t},$$

where the last inequality is due to the parameter bound in Lemma 19. $\qquad\square$

**Lemma 21** (Modified descent lemma). *Consider a loss satisfying 5A. Suppose there exists $s < T$ such that*

$$L(\mathbf{w}_s) \leq \frac{1}{4C_\beta^2 \eta},$$

*then for every $t \in [s, T]$ we have*

1. $L(\mathbf{w}_t) \leq 1/(4C_\beta^2 \eta)$ and $G(\mathbf{w}_t) \leq 1/(4C_\beta \eta)$,

2. $L(\mathbf{w}_{t+1}) \leq L(\mathbf{w}_t) - \frac{3\eta}{4}\|\nabla L(\mathbf{w}_t)\|^2 \leq L(\mathbf{w}_t)$,

*where $G(\mathbf{w})$ is defined in* (6).

*Proof.* We first show that Claim 1 implies Claim 2. By Assumption 5A, we have

$$\ell(\langle \mathbf{w}_{t+1}, \mathbf{z}_i \rangle) \leq \ell(\langle \mathbf{w}_t, \mathbf{z}_i \rangle) + \ell'(\langle \mathbf{w}_t, \mathbf{z}_i \rangle)\langle \mathbf{w}_{t+1} - \mathbf{w}_t, \mathbf{z}_i \rangle + C_\beta g(\langle \mathbf{w}_t, \mathbf{z}_i \rangle)\langle \mathbf{w}_{t+1} - \mathbf{w}_t, \mathbf{z}_i \rangle^2$$
$$\leq \ell(\langle \mathbf{w}_t, \mathbf{z}_i \rangle) + \ell'(\langle \mathbf{w}_t, \mathbf{z}_i \rangle)\langle \mathbf{w}_{t+1} - \mathbf{w}_t, \mathbf{z}_i \rangle + C_\beta g(\langle \mathbf{w}_t, \mathbf{z}_i \rangle)\|\mathbf{w}_{t+1} - \mathbf{w}_t\|^2.$$

Taking average over $i \in [n]$, we get

$$\begin{aligned}
L(\mathbf{w}_{t+1}) &\leq L(\mathbf{w}_t) + \langle \nabla L(\mathbf{w}_t), \mathbf{w}_{t+1} - \mathbf{w}_t \rangle + C_\beta G(\mathbf{w}_t)\|\mathbf{w}_{t+1} - \mathbf{w}_t\|^2 \\
&= L(\mathbf{w}_t) - \eta\|\nabla L(\mathbf{w}_t)\|^2 + C_\beta \eta^2 G(\mathbf{w}_t)\|\nabla L(\mathbf{w}_t)\|^2 \\
&\leq L(\mathbf{w}_t) - \eta\|\nabla L(\mathbf{w}_t)\|^2 + \frac{\eta}{4}\|\nabla L(\mathbf{w}_t)\|^2 \qquad\qquad \text{(Claim 1)} \\
&= L(\mathbf{w}_t) - \frac{3\eta}{4}\|\nabla L(\mathbf{w}_t)\|^2,
\end{aligned}$$

which verifies Claim 2.

Next, we prove Claim 1 by induction. The base case $t = s$ holds by Assumption 5A as follows:

$$G(\mathbf{w}_s) = \frac{1}{n}\sum_{i=1}^n g(\langle \mathbf{w}_s, \mathbf{z}_i \rangle) \leq \frac{1}{n}\sum_{i=1}^n C_\beta \ell(\langle \mathbf{w}_s, \mathbf{z}_i \rangle) = C_\beta L(\mathbf{w}_s) \leq \frac{1}{4C_\beta \eta}. \qquad (14)$$

To prove the step case, we suppose $L(\mathbf{w}_k) \leq 1/(4C_\beta^2 \eta)$ and $G(\mathbf{w}_k) \leq 1/(4C_\beta \eta)$ for $k = s, s + 1, \ldots, t$ and prove them for $k = t + 1$. Since Claim 1 implies Claim 2, we have

$$L(\mathbf{w}_{t+1}) \leq L(\mathbf{w}_t) \leq \cdots \leq L(\mathbf{w}_s) \leq \frac{1}{4C_\beta^2 \eta}.$$

Since $G(\mathbf{w}_{t+1}) \leq C_\beta L(\mathbf{w}_{t+1})$ holds as in (14), we have

$$G(\mathbf{w}_{t+1}) \leq C_\beta L(\mathbf{w}_{t+1}) \leq \frac{1}{4C_\beta \eta}.$$

Thus, the step case is shown, and all claims are proven. $\qquad\square$

**Lemma 22.** *Consider a nonincreasing and nonnegative loss $\ell$. For every $\mathbf{w}$ such that*

$$L(\mathbf{w}) \leq \frac{\ell(0)}{n},$$

*we have $y_i \langle \mathbf{w}, \mathbf{x}_i \rangle \geq 0$ for $i \in [n]$.*

*Proof.* See [63, Lemma 31]. $\qquad\square$

**Lemma 23** (Stable phase). *Consider a nonincreasing loss $\ell$ satisfying Assumptions 5A and 3B. Suppose there exists $s < T$ such that*

$$L(\mathbf{w}_s) \leq \min\left\{\frac{1}{4C_\beta^2 \eta}, \frac{\ell(0)}{n}\right\}.$$

*Then, for every $t \in [0, T - s]$, we have*

$$L(\mathbf{w}_{s+t}) \leq 5\frac{\rho(\gamma^2 \eta t)}{\gamma^2 \eta t}.$$

*Proof.* By the lemma assumption, we can apply Lemma 21 for $s$ onwards. Therefore, we have for $k \geq 0$,

$$\eta \|\nabla L(\mathbf{w}_{s+k})\|^2 \leq \frac{4}{3}[L(\mathbf{w}_{s+k}) - L(\mathbf{w}_{s+k+1})] \leq \frac{4}{3}L(\mathbf{w}_{s+k}). \tag{15}$$

Choose a comparator centered at $\mathbf{w}_s$,

$$\mathbf{u} := \mathbf{w}_s + \theta \mathbf{w}_*, \quad \theta := \frac{\sqrt{\rho(\eta^2\gamma t)}}{\gamma}.$$

For $k \leq t - 1$, we have

$$\|\mathbf{w}_{s+k+1} - \mathbf{u}\|^2 = \|\mathbf{w}_{s+k} - \mathbf{u}\|^2 + 2\eta \langle \nabla L(\mathbf{w}_{s+k}), \mathbf{u} - \mathbf{w}_{s+k}\rangle + \eta^2\|\nabla L(\mathbf{w}_{s+k})\|^2$$

$$\leq \|\mathbf{w}_{s+k} - \mathbf{u}\|^2 + 2\eta \langle \nabla L(\mathbf{w}_{s+k}), \mathbf{u} - \mathbf{w}_{s+k}\rangle + \frac{4}{3}\eta L(\mathbf{w}_{s+k}). \quad \text{(by (15))}$$

Following the same derivation of (7), we can bound the second term as follows:

$$\langle \nabla L(\mathbf{w}_{s+k}), \mathbf{u} - \mathbf{w}_{s+k}\rangle \leq \frac{1}{n}\sum_{i=1}^{n} \ell(\theta\gamma + \langle \mathbf{w}_s, \mathbf{z}_i\rangle) - L(\mathbf{w}_{s+k}).$$

The assumption $L(\mathbf{w}_s) \leq \ell(0)/n$ allows us to apply Lemma 22, so $\langle \mathbf{w}_s, \mathbf{z}_i\rangle \geq 0$ and thus

$$\ell(\theta\gamma + \langle \mathbf{w}_s, \mathbf{z}_i\rangle) \leq \ell(\theta\gamma) = \ell(\sqrt{\rho(\gamma^2\eta t)}),$$

where $\ell$ is nonincreasing due to the lemma assumption. Consequently, we can control the second term by

$$\langle \nabla L(\mathbf{w}_{s+k}), \mathbf{u} - \mathbf{w}_{s+k}\rangle \leq \ell(\sqrt{\rho(\gamma^2\eta t)}) - L(\mathbf{w}_{s+k}).$$

Plugging this back, we get

$$\|\mathbf{w}_{s+k+1} - \mathbf{u}\|^2 \leq \|\mathbf{w}_{s+k} - \mathbf{u}\|^2 + 2\eta[\ell(\sqrt{\rho(\gamma^2\eta t)}) - L(\mathbf{w}_{s+k})] + \frac{4}{3}\eta L(\mathbf{w}_{s+k})$$

$$\leq \|\mathbf{w}_{s+k} - \mathbf{u}\|^2 + 2\eta\ell(\sqrt{\rho(\gamma^2\eta t)}) - \frac{2}{3}\eta L(\mathbf{w}_{s+k}).$$

Telescoping the sum from $0$ to $t - 1$ and rearranging, we get

$$\frac{3\|\mathbf{w}_{s+t} - \mathbf{u}\|^2}{2\eta t} + \frac{1}{t}\sum_{k=0}^{t-1} L(\mathbf{w}_{s+k}) \leq 3\ell(\sqrt{\rho(\gamma^2\eta t)}) + \frac{3\|\mathbf{w}_s - \mathbf{u}\|^2}{2\eta t}$$

$$\leq 3\frac{\rho(\gamma^2\eta t)}{\gamma^2\eta t} + \frac{3\|\mathbf{w}_s - \mathbf{u}\|^2}{2\eta t}. \quad \text{(Lemma 7)}$$

Finally, we can show the claims as follows:

$$\frac{1}{t}\sum_{k=0}^{t-1} L(\mathbf{w}_{s+k}) \leq 3\frac{\rho(\gamma^2\eta t)}{\gamma^2\eta t} + \frac{3\|\mathbf{w}_s - \mathbf{u}\|^2}{2\eta t}$$

$$= \frac{9}{2}\frac{\rho(\gamma^2\eta t)}{\gamma^2\eta t}$$

$$\leq 5\frac{\rho(\gamma^2\eta t)}{\gamma^2\eta t}.$$

By Lemma 21, $L(\mathbf{w}_t)$ is nonincreasing for $t \geq s$, and thus we have

$$L(\mathbf{w}_{s+t}) \leq \frac{1}{t}\sum_{k=0}^{t-1} L(\mathbf{w}_{s+k}) \leq 5\frac{\rho(\gamma^2\eta t)}{\gamma^2\eta t}.$$

$\square$

**Lemma 24** (Phase transition). *Suppose Assumption 2 and consider a convex and nonincreasing loss $\ell$ that satisfies Assumptions 3B and 3C. Define*

$$\psi(\lambda) = \frac{\lambda}{\rho(\lambda)}, \quad \lambda > 0.$$

*Then, there is $C > 0$ depending on $C_g$, $C_\beta$, and $\ell(0)$ such that the following holds. Let*

$$\tau := \frac{1}{\gamma^2} \max\left\{ \frac{\psi^{-1}(C(\eta+n))}{\eta}, C(\eta+n)\eta \right\}.$$

*If $\tau < T$, then there exists $s \in [0, \tau]$ such that*

$$L(\mathbf{w}_s) \leq \min\left\{ \frac{1}{4C_\beta^2\eta}, \frac{\ell(0)}{n} \right\}.$$

*Proof.* Applying Lemma 19 with $t = \tau$, we have

$$\frac{1}{\tau} \sum_{k=0}^{\tau-1} L(\mathbf{w}_k) \leq \frac{[6\sqrt{\rho(\gamma^2\eta\tau)} + \eta C_g]^2}{8\gamma^2\eta\tau} = \left[ \frac{3}{\sqrt{2}} \sqrt{\frac{\rho(\gamma^2\eta\tau)}{\gamma^2\eta\tau}} + \frac{\sqrt{2}}{4} \frac{\eta C_g}{\sqrt{\gamma^2\eta\tau}} \right]^2.$$

Choose $\tau$ such that

$$\gamma^2\eta\tau \geq \max\left\{ \psi^{-1}\left( 18[4C_\beta^2\eta + n/\ell(0)] \right), \frac{1}{2}(\eta C_g)^2(4C_\beta^2\eta + n/\ell(0)) \right\}.$$

It is clear that

$$\frac{1}{\psi(\lambda)} = \frac{\rho(\lambda)}{\lambda} = \min_z \ell(z) + \frac{z^2}{\lambda}$$

is a decreasing function. Then, we have

$$\frac{3}{\sqrt{2}} \sqrt{\frac{\rho(\gamma^2\eta\tau)}{\gamma^2\eta\tau}} = \frac{3}{\sqrt{2}} \sqrt{\frac{1}{\psi(\gamma^2\eta\tau)}} \leq \frac{3}{\sqrt{2}} \sqrt{\frac{1}{18[4C_\beta^2\eta + n/\ell(0)]}} = \frac{1}{2} \frac{1}{\sqrt{4C_\beta^2\eta + n/\ell(0)}}$$

and

$$\frac{\sqrt{2}}{4} \frac{\eta C_g}{\sqrt{\gamma^2\eta\tau}} \leq \frac{1}{2} \frac{1}{\sqrt{4C_\beta^2\eta + n/\ell(0)}}.$$

These two inequalities together imply that

$$\frac{1}{\tau} \sum_{k=0}^{\tau-1} L(\mathbf{w}_k) \leq \frac{1}{4C_\beta^2\eta + n/\ell(0)} \leq \min\left\{ \frac{1}{4C_\beta^2\eta}, \frac{\ell(0)}{n} \right\},$$

which implies that there exists $s \leq \tau$ for $L(\mathbf{w}_s)$ satisfies the right-hand side bound. $\square$

**Lemma 25** (Phase transition time under exponential tail). *Suppose Assumption 2 and consider a nonincreasing loss $\ell$ satisfying Assumptions 3B and 3C, and 5B. Furthermore, assume $\eta \geq 1$. Then, there exists $C > 0$ depending on $C_e$, $C_g$, $C_\beta$, $\ell(0)$, and $n$ such that the following holds. Let*

$$\tau := \frac{C}{\gamma^2} \max\left\{ \eta, n \ln n \right\}.$$

*If $\tau \leq T$, then there exists $s \in [0, \tau]$ such that*

$$L(\mathbf{w}_s) \leq \min\left\{ \frac{1}{4C_\beta^2\eta}, \frac{\ell(0)}{n} \right\}.$$

*Proof.* Under Assumption 5B, we have

$$\ell(z) \leq C_e g(z) = -C_e \ell'(z), \quad \text{for } z \geq 0,$$

which implies

$$\frac{\ell'(z)}{\ell(z)} \leq -C_e^{-1}, \quad \text{for } z \geq 0.$$

Integrating both sides, we get

$$\ln \ell(z) \leq \ln \ell(0) + \int_0^z \frac{\ell'(\zeta)}{\ell(\zeta)} d\zeta \leq \ln \ell(0) - C_e^{-1} z, \quad \text{for } z \geq 0,$$

which implies

$$\ell(z) \leq \ell(0) \exp(-C_e^{-1} z), \quad \text{for } z \geq 0.$$

Using the exponential tail property, we have

$$\rho(\lambda) = \min_{z \in \mathbb{R}} \lambda \ell(z) + z^2 \leq \lambda \ell(C_e \ln(\lambda)) + C_e^2 \ln^2(\lambda) \leq \ell(0) + C_e^2 \ln^2(\lambda).$$

Applying Lemma 20, we have

$$\frac{1}{\tau} \sum_{k=0}^{\tau-1} G(\mathbf{w}_k) \leq \frac{4\sqrt{\rho(\gamma^2 \eta \tau)} + \eta C_g}{\gamma^2 \eta \tau}$$

$$\leq \frac{4\sqrt{\ell(0) + C_e^2 \ln^2(\gamma^2 \eta \tau)} + \eta C_g}{\gamma^2 \eta \tau}$$

$$\leq \frac{4\sqrt{\ell(0)} + 4C_e \ln(\gamma^2 \eta \tau) + \eta C_g}{\gamma^2 \eta \tau} \qquad (\sqrt{a+b} \leq \sqrt{a} + \sqrt{b})$$

$$\leq \frac{4C_e}{\eta} \frac{\ln(\gamma^2 \tau)}{\gamma^2 \tau} + \frac{C_g + 4C_e}{\gamma^2 \tau} + \frac{4\sqrt{\ell(0)}}{\eta} \frac{1}{\gamma^2 \tau}.$$

Here, we take $C > 0$ depending on $C_e$, $C_g$, $C_\beta$, $\ell(0)$, and additionally $n$ such that

$$\gamma^2 \tau \geq C \max\{\eta, n\}$$

and

$$\frac{\ln C}{C} \leq \frac{\min\left\{\frac{1}{4C_e C_\beta^2}, \frac{\ell(0)}{C_e}\right\}}{4C_e(1 + \ln n) + C_g + 4C_e + 4\sqrt{\ell(0)}}.$$

This choice is possible with sufficiently large $C \geq e$ because $(\ln C)/C$ is strictly decreasing in $C \geq e$ toward zero. Such $C$ enables us to have

$$\frac{1}{\tau} \sum_{k=0}^{\tau-1} G(\mathbf{w}_k)$$

$$\leq \frac{1}{C \max\{\eta, n\}} \left[ \frac{4C_e}{\eta}(\ln C + \ln \max\{\eta, n\}) + C_g + 4C_e + \frac{4\sqrt{\ell(0)}}{\eta} \right]$$

$$\leq \frac{4C_e(\ln C + \ln n) + C_g + 4C_e + 4\sqrt{\ell(0)}}{C \max\{\eta, n\}} \qquad (\eta \geq 1)$$

$$\leq \frac{\ln C}{C} \frac{4C_e(1 + \ln n) + C_g + 4C_e + 4\sqrt{\ell(0)}}{\max\{\eta, n\}} \qquad (\ln C \geq 1)$$

$$\leq \frac{\min\left\{\frac{1}{4C_e C_\beta^2}, \frac{\ell(0)}{C_e}\right\}}{\max\{\eta, n\}}$$

$$\leq \min\left\{ \frac{1}{4C_e C_\beta^2 \eta}, \frac{\ell(0)}{C_e n} \right\}.$$

From this we have some $s \leq \tau$ such that

$$G(\mathbf{w}_s) \leq \min\left\{\frac{1}{4C_eC_\beta^2\eta}, \frac{\ell(0)}{C_e n}\right\}.$$

This ensures that for every $i \in [n]$,

$$\frac{1}{n}g(\langle\mathbf{w}_s, \mathbf{z}_i\rangle) \leq G(\mathbf{w}_s) = \frac{1}{n}\sum_{i=1}^{n}g(\langle\mathbf{w}_s, \mathbf{z}_i\rangle) \leq \frac{\ell(0)}{C_e n} \leq \frac{g(0)}{n},$$

where the last inequality is due to Assumption 5B. The above implies $\langle\mathbf{w}_s, \mathbf{z}_i\rangle \geq 0$ since $g(\cdot)$ is nonincreasing. Thus, we can apply Assumption 5B for $\langle\mathbf{w}_s, \mathbf{z}_i\rangle$ and get

$$\ell(\langle\mathbf{w}_s, \mathbf{z}_i\rangle) \leq C_eg(\langle\mathbf{w}_s, \mathbf{z}_i\rangle).$$

Taking an average over $i \in [n]$, we have

$$L(\mathbf{w}_s) \leq C_eG(\mathbf{w}_s).$$

We complete the proof by plugging in the upper bound on $G(\mathbf{w}_s)$. □

# E    Separation margin and self-bounding property

In this section, we discuss the relationship between separation margin and the self-bounding property. First, we show that a loss function does not have separation margin if it satisfies the self-bounding property.

**Proposition 26.** *Consider a loss $\ell \colon \mathbb{R} \to \mathbb{R}$ that is continuously differentiable and nonincreasing, and satisfies $\ell(z_0) > 0$ for some $z_0 \in \mathbb{R}$. If $\ell$ satisfies Assumption 5A, then $\ell$ does not have separation margin.*

*Proof.* Choose any $\varepsilon \in (0, 1/C_\beta)$. The convexity of $\ell$ implies that

$$g(z) = -\ell'(z) \geq \frac{\ell(z) - \ell(z + \varepsilon)}{\varepsilon} \quad \text{for any } z \in \mathbb{R}.$$

By the self-bounding property (Assumption 5A), we further have

$$\frac{\ell(z) - \ell(z + \varepsilon)}{\varepsilon} \leq g(z) \leq C_\beta\ell(z).$$

Solving this, we have

$$\ell(z + \varepsilon) \geq (1 - C_\beta\varepsilon)\ell(z).$$

Thus, if $\ell(z) > 0$ holds, we additionally have $\ell(z + \varepsilon) > 0$ for $\varepsilon \in (0, 1/C_\beta)$, and we conclude that $\ell$ cannot have separation margin because $\ell > 0$ holds on the entire $\mathbb{R}$. □

Next, we argue that the converse of Proposition 26 does not hold, that is, even if a loss $\ell$ does not have separation margin, it does not always imply that $\ell$ satisfies the self-bounding property. A counterexample is a Fenchel–Young loss generated by the following potential function:

$$\phi(\mu) = \int_0^\mu \Phi^{-1}(p)\mathrm{d}p, \quad \text{where } \Phi \text{ is the standard normal CDF } \Phi(x) := \frac{1}{2}\left[1 + \text{erf}\left(\frac{x}{\sqrt{2}}\right)\right]$$

and erf is the error function. The generated Fenchel–Young loss is relevant to the probit model [40] because $\phi'$ is nothing else but the probit link function prevailing in generalized linear models. Hence, we call the generated Fenchel–Young loss the *probit Fenchel–Young loss* for convenience. We can have a concise form of the probit Fenchel-Young loss:

$$\ell(z) = \phi^*(-z)$$

$$= \int_{-\infty}^{-z}(\phi')^{-1}(\zeta)\mathrm{d}\zeta$$

$$= \int_{-\infty}^{-z}\Phi(\zeta)\mathrm{d}\zeta$$

$$= [\zeta\Phi(\zeta) + \Phi'(\zeta)]_{-\infty}^{-z}$$
$$= -z\Phi(-z) + \Phi'(-z).$$

The probit Fenchel–Young loss does not have separation margin because $\phi'(\mu) = \Phi^{-1}(\mu) \to -\infty$ as $\mu \downarrow 0$ (see Proposition 4). However, it does not satisfy the self-bounding property. To see this, we have

$$\frac{g(z)}{\ell(z)} = -\frac{\ell'(z)}{\ell(z)}$$

$$= -\frac{-\Phi(-z)}{-z\Phi(-z) + \Phi'(-z)}$$

$$= \frac{\Phi(\bar{z})}{\bar{z}\Phi(\bar{z}) + \Phi'(\bar{z})}, \qquad\qquad (\bar{z} \equiv -z)$$

which implies

$$\lim_{z\to\infty} \frac{g(z)}{\ell(z)} = \lim_{\bar{z}\to-\infty} \frac{\Phi'(\bar{z})}{\Phi(\bar{z}) + \bar{z}\Phi'(\bar{z}) + \Phi''(\bar{z})} \qquad \text{(L'Hôpital's rule)}$$

$$= \lim_{\bar{z}\to-\infty} \frac{\Phi'(\bar{z})}{\Phi(\bar{z})}$$

$$= \lim_{\bar{z}\to-\infty} \frac{\Phi''(\bar{z})}{\Phi'(\bar{z})} \qquad \text{(L'Hôpital's rule)}$$

$$= \lim_{\bar{z}\to-\infty} \frac{-\bar{z}\Phi'(\bar{z})}{\Phi'(\bar{z})}$$

$$= \infty.$$

Hence, $g(z)$ cannot always be bounded from above by $\ell(z)$, that is, the self-bounding property is not satisfied.

## F   Omitted calculation for examples

Here, we compute for each $\phi$,
$$\lim_{\mu\downarrow0} \frac{\phi'(\mu)}{\mu\phi''(\mu)} \left[1 - \frac{\phi(\mu)}{\mu\phi'(\mu)}\right]$$
to estimate the power $\alpha$ of the convergence rate provided in Theorem 5, by making the error parameter $\bar{\varepsilon} > 0$ in (3) arbitrarily small. Correspondingly, we compute

$$\lim_{\mu\downarrow0} \frac{\mu}{[\mu\phi'(\mu) - \phi(\mu)]^{\alpha}}$$

to estimate the constant $C_\phi$ in the convergence rate, verifying that $C_\phi$ neither degenerates nor diverges for arbitrarily small error parameter $\bar{\varepsilon} > 0$.

Before proceeding with each example, we provide a rough estimate of $\rho$ for loss functions without separation margin.

**Lemma 27.** *Consider a loss $\ell$ satisfying Assumptions 3A and 3B that does not have separation margin. Then,*

$$\rho(\lambda) \le -\phi\left(\frac{1}{2}\right)\lambda.$$

*Proof.* First, we rewrite $\rho$ as a dual form. By introducing the dual variable $\mu$ of $z$ by

$$z = \phi'(\mu) \quad \text{and} \quad \mu = (\phi^*)'(z),$$

we have

$$\rho(\lambda) = \min_{z\in\mathbb{R}} \lambda\ell(z) + z^2$$
$$= \min_{z\in\mathbb{R}} \lambda\phi^*(z) + z^2$$

$$= \min_{\mu \in [0,1]} \lambda[\mu\phi'(\mu) - \phi(\mu)] + [\phi'(\mu)]^2,$$

where we use the definition of the convex conjugate $\phi^*(z) = \mu z - \phi(\mu)$ at the last identity. Now, we write the objective as $R(\mu)$:

$$R(\mu) := \lambda[\mu\phi'(\mu) - \phi(\mu)] + [\phi'(\mu)]^2.$$

Differentiating $R$, we have

$$R'(\mu_\star) = [\lambda\mu_\star + 2\phi'(\mu_\star)]\phi''(\mu_\star) = 0 \quad \overset{\phi'' \geq 0}{\Longrightarrow} \quad \phi'(\mu_\star) = -\frac{\lambda}{2}\mu_\star$$

at the minimizer $\mu_\star$ of $R$. Plugging this back to $R$, we have

$$\rho(\lambda) = R(\mu_\star) = \lambda\left[\mu_\star\left(-\frac{\lambda}{2}\mu_\star\right) - \phi(\mu_\star)\right] + \left(-\frac{\lambda}{2}\mu_\star\right)^2 = -\lambda\phi(\mu_\star) \leq -\phi\left(\frac{1}{2}\right)\lambda,$$

where the last inequality owes to that a convex potential satisfying Assumption 1 is minimized at the uniform distribution $\mu_\star = 1/2$ [11, Proposition 4]. □

By using Lemma 27, we can simplify the convergence rate of (GD) given by Theorem 5 for a loss that does not have separation margin. Note that the following convergence rate is not sufficiently tight due to overestimation of $\rho$ by Lemma 27; nevertheless, the provided convergence rate is convenient when we do not have an access to $\rho$ analytically.

**Corollary 28.** *Under the same setup with Theorem 5, we additionally assume that $\ell$ does not have separation margin. If $(\alpha, C_\phi)$ with (3) satisfies $\alpha, C_\phi \in (0, \infty)$ and*

$$T > \frac{2C_g n}{C_\phi \gamma^2}\varepsilon^{-\alpha} + \frac{16[-\phi(1/2)]n^2}{C_\phi^2 \gamma^2 \eta}\varepsilon^{-2\alpha} \quad \text{for } \varepsilon \in (0, \bar{\varepsilon}),$$

*then we have $L(\mathbf{w}_T) \leq \varepsilon$.*

*Proof.* Combining Theorem 5 and Lemma 27, we have the following convergence rate:

$$T > \frac{4n\sqrt{-\phi(1/2)}\varepsilon^{-\alpha}}{C_\phi\gamma\sqrt{\eta}}\sqrt{T} + \frac{C_g n\varepsilon^{-\alpha}}{C_\phi\gamma^2}.$$

Defining

$$a := \frac{4n\sqrt{-\phi(1/2)}\varepsilon^{-\alpha}}{C_\phi\gamma\sqrt{\eta}} \quad \text{and} \quad b := \frac{C_g n\varepsilon^{-\alpha}}{C_\phi\gamma^2},$$

we have the following inequality in $T$:

$$T^2 - (a^2 + 2b)T + b^2 > 0.$$

This can be solved for $T \geq 1$:

$$T > \frac{a^2 + 2b}{2}\left[1 + \underbrace{\sqrt{1 - \left(\frac{2b}{a^2 + 2b}\right)^2}}_{\leq 1}\right],$$

for which $T > a^2 + 2b$ is sufficient. Thus, we have shown the statement. □

Throughout this section, we repeatedly use L'Hôpital's rule. When it is used, we notate by (‡).

### F.1 Shannon entropy

For the Shannon entropy $\phi(\mu) = \mu \ln \mu + (1 - \mu) \ln(1 - \mu)$, we have

$$\phi'(\mu) = \ln \mu - \ln(1 - \mu) \quad \text{and} \quad \phi''(\mu) = \frac{1}{\mu} + \frac{1}{1 - \mu},$$

which imply

$$\lim_{\mu \downarrow 0} \frac{\phi'(\mu)}{\mu \phi''(\mu)} \left[ 1 - \frac{\phi(\mu)}{\mu \phi'(\mu)} \right] = \lim_{\mu \downarrow 0} \frac{\ln \frac{\mu}{1 - \mu}}{1 - \frac{\mu}{1 - \mu}} \left[ 1 - \frac{\mu \ln \mu + (1 - \mu) \ln(1 - \mu)}{\mu \ln \mu - \mu \ln(1 - \mu)} \right]$$

$$= \lim_{\mu \downarrow 0} \ln \frac{\mu}{1 - \mu} \cdot \frac{\mu \ln \mu - \mu \ln(1 - \mu) - \mu \ln \mu - (1 - \mu) \ln(1 - \mu)}{\mu \ln \frac{\mu}{1 - \mu}}$$

$$= \lim_{\mu \downarrow 0} \frac{-\ln(1 - \mu)}{\mu}$$

$$\overset{(\ddagger)}{=} \lim_{\mu \downarrow 0} \frac{1}{1 - \mu}$$

$$= 1,$$

and

$$\lim_{\mu \downarrow 0} \frac{\mu}{\mu \phi'(\mu) - \phi(\mu)} = \lim_{\mu \downarrow 0} \frac{\mu}{\mu \ln \mu - \mu \ln(1 - \mu) - \mu \ln \mu - (1 - \mu) \ln(1 - \mu)}$$

$$= \lim_{\mu \downarrow 0} \frac{\mu}{-\ln(1 - \mu)} \cdot \frac{1}{2\mu - 1}$$

$$= \lim_{\mu \downarrow 0} \frac{\mu}{\ln(1 - \mu)}$$

$$\overset{(\ddagger)}{=} \lim_{\mu \downarrow 0} (1 - \mu)$$

$$= 1.$$

Finally, we derive the convergence rate for the logistic loss. Plugging $\alpha = 1$, $C_\phi = 1$, $C_g = 1$, and $\rho(\lambda) \le 1 + \ln^2(\lambda) \le 2 \ln^2(\lambda)$ to Theorem 5, we have

$$T > \frac{n}{\gamma^2} \left( \frac{4\sqrt{2} \ln(\gamma^2 \eta T)}{\eta} + 1 \right) \varepsilon^{-1} = \left[ \frac{4\sqrt{2} \ln(\gamma^2 \eta)}{\eta} + 1 + \frac{4\sqrt{2}}{\eta} \ln T \right] \frac{n \varepsilon^{-1}}{\gamma^2}.$$

Dividing both ends by $\ln T$, we have

$$\frac{T}{\ln T} > \left[ \left( \frac{4\sqrt{2} \ln(\gamma^2 \eta)}{\eta} + 1 \right) \frac{1}{\ln T} + \frac{4\sqrt{2}}{\eta} \right] \frac{n \varepsilon^{-1}}{\gamma^2},$$

for which the following is sufficient when $T \ge 2$:

$$\frac{T}{\ln T} > \left[ \left( \frac{4\sqrt{2} \ln(\gamma^2 \eta)}{\eta} + 1 \right) \frac{1}{\ln 2} + \frac{4\sqrt{2}}{\eta} \right] \frac{n \varepsilon^{-1}}{\gamma^2}$$

$$= \left[ \frac{4\sqrt{2} \log_2(\gamma^2 \eta)}{\eta} + \frac{1}{\ln 2} + \frac{4\sqrt{2}}{\eta} \right] \frac{n \varepsilon^{-1}}{\gamma^2}.$$

By ignoring the logarithmic factor, we have

$$T \gtrsim \left[ \frac{4\sqrt{2} \log_2(\gamma^2 \eta)}{\eta} + \frac{1}{\ln 2} + \frac{4\sqrt{2}}{\eta} \right] \frac{n \varepsilon^{-1}}{\gamma^2}.$$

### F.2 Semi-circle entropy

For the semi-circle entropy $\phi(\mu) = -2\sqrt{\mu(1-\mu)}$, we first derive the analytical form of the corresponding Fenchel–Young loss. We have

$$\phi'(\mu) = \frac{2\mu - 1}{\sqrt{\mu(1-\mu)}} \quad \text{and} \quad \phi''(\mu) = \frac{1}{2[\mu(1-\mu)]^{3/2}}.$$

The dual transform $(\phi^*)'$ is given by

$$(\phi^*)'(z) = (\phi')^{-1}(z) = \frac{1}{2}\left[\frac{\frac{z}{2}}{\sqrt{\left(\frac{z}{2}\right)^2 + 1}} + 1\right],$$

thanks to the Danskin's theorem [21]. Then, we can derive the Fenchel–Young loss by the definition of the convex conjugate:

$$\ell(z) = \phi^*(-z) = -z(\phi^*)'(z) - \phi\left((\phi^*)'(-z)\right) = \frac{-z + \sqrt{z^2 + 4}}{2}.$$

Next, we compute the loss parameters $\alpha$ and $C_\phi$ respectively as follows:

$$\lim_{\mu \downarrow 0} \frac{\phi'(\mu)}{\mu \phi''(\mu)}\left[1 - \frac{\phi(\mu)}{\mu \phi'(\mu)}\right] = \lim_{\mu \downarrow 0} \frac{\frac{2\mu-1}{\sqrt{\mu(1-\mu)}}}{\frac{\mu}{2[\mu(1-\mu)]^{3/2}}}\left[1 + \frac{2\sqrt{\mu(1-\mu)}}{\frac{\mu(2\mu-1)}{\sqrt{\mu(1-\mu)}}}\right]$$

$$= \lim_{\mu \downarrow 0} 2(2\mu - 1)(1 - \mu)\left[1 + \frac{2(1-\mu)}{2\mu - 1}\right]$$

$$= 2,$$

and

$$\lim_{\mu \downarrow 0} \frac{\mu}{[\mu\phi'(\mu) - \phi(\mu)]^2} = \lim_{\mu \downarrow 0} \frac{\mu}{\left[\mu\frac{2\mu-1}{\sqrt{\mu(1-\mu)}} + 2\sqrt{\mu(1-\mu)}\right]^2}$$

$$= \lim_{\mu \downarrow 0}(1 - \mu)$$

$$= 1.$$

To estimate $\rho(\lambda)$,

$$\rho(\lambda) = \min_{z \in \mathbb{R}} \lambda\ell(z) + z^2$$

$$\leq \lambda\frac{-\ln\lambda + \sqrt{\ln^2\lambda + 4}}{2} + \ln^2\lambda \qquad\qquad (z = \ln\lambda)$$

$$= \frac{2\lambda}{\ln\lambda + \sqrt{\ln^2\lambda + 4}} + \ln^2\lambda$$

$$\leq \frac{5\lambda}{2\ln\lambda},$$

where we used

$$\frac{\lambda}{\ln\lambda} \geq \frac{2\lambda}{\ln\lambda + \sqrt{\ln^2\lambda + 4}} \geq \frac{2}{3}\cdot\ln^2\lambda \quad \text{for } \lambda \geq 1.$$

Finally, we derive the convergence rate for the semi-circle loss. Plugging $\alpha = 2$, $C_\phi = 1$, $C_g = 1$, and $\rho(\lambda) \leq 5\lambda/(2\ln\lambda)$ to Theorem 5, we have

$$T > \frac{n}{\gamma^2}\left(\frac{4\sqrt{\frac{5}{2}\frac{\gamma^2\eta T}{\ln(\gamma^2\eta T)}}}{\eta} + 1\right)\varepsilon^{-2} = \left[\frac{2\sqrt{10}}{\eta}\sqrt{\frac{\gamma^2\eta T}{\ln(\gamma^2\eta T)}} + 1\right]\frac{n\varepsilon^{-2}}{\gamma^2},$$

for which the following is sufficient when $T \geq 2$:

$$T > \left\lceil \frac{2\sqrt{10}}{\eta} \sqrt{\frac{\gamma^2 \eta T}{\ln(2\gamma^2 \eta)}} + 1 \right\rceil \frac{n\varepsilon^{-2}}{\gamma^2}.$$

Subsequently, we follow the same flow as in the proof of Corollary 28. Defining

$$a := \frac{2\sqrt{10}n\varepsilon^{-2}}{\gamma^2 \eta} \sqrt{\frac{\gamma^2 \eta}{\ln(2\gamma^2 \eta)}} \quad \text{and} \quad b := \frac{n\varepsilon^{-2}}{\gamma^2},$$

we have the following inequality in $T$:

$$T^2 - (a^2 + 2b)T + b^2 > 0.$$

This can be solved for $T \geq 1$:

$$T > \frac{a^2 + 2b}{2} \left[ 1 + \underbrace{\sqrt{1 - \left(\frac{2b}{a^2 + 2b}\right)^2}}_{\leq 1} \right],$$

for which $T > a^2 + 2b$ is sufficient, namely,

$$T > \frac{40n^2}{\gamma^2 \eta \ln(2\gamma^2 \eta)} \varepsilon^{-4} + \frac{2n}{\gamma^2} \varepsilon^{-2}$$

is sufficient. Thus, the convergence rate is $T = \Omega(\varepsilon^{-4})$.

### F.3  Tsallis entropy

For the Tsallis entropy

$$\phi(\mu) = \frac{\mu^q + (1-\mu)^q - 1}{q - 1},$$

define

$$\phi_0(\mu) = \mu^q + (1-\mu)^q - 1,$$
$$\phi_1(\mu) = \mu^{q-1} - (1-\mu)^{q-1},$$
$$\phi_2(\mu) = \mu^{q-2} + (1-\mu)^{q-2}.$$

When $0 < q < 2$ ($q \neq 1$),

$$\lim_{\mu \downarrow 0} \frac{\phi'(\mu)}{\mu \phi''(\mu)} \left[ 1 - \frac{\phi(\mu)}{\mu \phi'(\mu)} \right] = \frac{1}{q(q-1)} \lim_{\mu \downarrow 0} \frac{1}{\phi_2(\mu)} \cdot \frac{q\mu\phi_1(\mu) - \phi_0(\mu)}{\mu^2}$$

$$= \frac{1}{q(q-1)} \lim_{\mu \downarrow 0} \frac{1}{1 + \left(\frac{\mu}{1-\mu}\right)^{2-q}} \cdot \frac{q\mu\phi_1(\mu) - \phi_0(\mu)}{\mu^q}$$

$$= \frac{1}{q(q-1)} \lim_{\mu \downarrow 0} \frac{q\mu\phi_1(\mu) - \phi_0(\mu)}{\mu^q}$$

$$\overset{(\ddagger)}{=} \frac{1}{q(q-1)} \lim_{\mu \downarrow 0} \frac{q\phi_1(\mu) + q(q-1)\mu\phi_2(\mu) - q\phi_1(\mu)}{q\mu^{q-1}}$$

$$= \frac{1}{q} \lim_{\mu \downarrow 0} \frac{\mu^{q-1} + (1-\mu)^{q-2}\mu}{\mu^{q-1}}$$

$$\overset{(\ddagger)}{=} \frac{1}{q} \lim_{\mu \downarrow 0} \frac{(q-1)\mu^{q-2} + (1-\mu)^{q-2} - (q-2)(1-\mu)^{q-3}\mu}{(q-1)\mu^{q-2}}$$

$$= \frac{1}{q} \lim_{\mu \downarrow 0} \left[ 1 + \frac{1}{q-1} \left(\frac{\mu}{1-\mu}\right)^{2-q} - (q-2)\left(\frac{\mu}{1-\mu}\right)^{3-q} \right]$$

$$= \frac{1}{q}.$$

In addition, we have

$$\lim_{\mu\downarrow0} \frac{\mu}{[\mu\phi'(\mu) - \phi(\mu)]^{1/q}}$$

$$= \lim_{\mu\downarrow0} \frac{(q-1)^{1/q}\mu}{[q\mu\phi_1(\mu) - \phi_0(\mu)]^{1/q}}$$

$$= (q-1)^{1/q} \left\{ \lim_{\mu\downarrow0} \frac{q\mu\phi_1(\mu) - \phi_0(\mu)}{\mu^q} \right\}^{-1/q}$$

$$= (q-1)^{1/q} \left\{ q - 1 - \lim_{\mu\downarrow0} \frac{q\mu(1-\mu)^{q-1} + (1-\mu)^q - 1}{\mu^q} \right\}^{-1/q}$$

$$\overset{(\ddagger)}{=} (q-1)^{1/q} \left\{ q - 1 - \lim_{\mu\downarrow0} \frac{q(1-\mu)^{q-1} - q(q-1)\mu(1-\mu)^{q-2} - q(1-\mu)^{q-1}}{q\mu^{q-1}} \right\}^{-1/q}$$

$$= (q-1)^{1/q} \left\{ q - 1 - (q-1)\lim_{\mu\downarrow0} \left(\frac{\mu}{1-\mu}\right)^{2-q} \right\}^{-1/q}$$

$$= (q-1)^{1/q} \cdot (q - 1 + 0)^{-1/q}$$

$$= 1.$$

When $q \geq 2$,

$$\lim_{\mu\downarrow0} \frac{\phi'(\mu)}{\mu\phi''(\mu)} \left[1 - \frac{\phi(\mu)}{\mu\phi'(\mu)}\right]$$

$$= \frac{1}{q(q-1)} \lim_{\mu\downarrow0} \frac{1}{\phi_2(\mu)} \cdot \frac{q\mu\phi_1(\mu) - \phi_0(\mu)}{\mu^2}$$

$$= \frac{1}{q(q-1)} \lim_{\mu\downarrow0} \frac{q\mu\phi_1(\mu) - \phi_0(\mu)}{\mu^2}$$

$$= \frac{1}{q(q-1)} \lim_{\mu\downarrow0} \frac{q[\mu^q - (1-\mu)^{q-1}\mu] - \mu^q - (1-\mu)^q}{\mu^2}$$

$$\overset{(\ddagger)}{=} \frac{1}{q(q-1)} \lim_{\mu\downarrow0} \frac{q[q\mu^{q-1} - (1-\mu)^{q-1} + (q-1)(1-\mu)^{q-2}\mu] - q\mu^{q-1} + q(1-\mu)^{q-1}}{2\mu}$$

$$= \lim_{\mu\downarrow0} \frac{\mu^{q-2} + (1-\mu)^{q-2}}{2}$$

$$= \frac{1}{2}.$$

In addition, we have

$$\lim_{\mu\downarrow0} \frac{\mu}{[\mu\phi'(\mu) - \phi(\mu)]^{1/2}} = (q-1)^{1/2} \left\{ \lim_{\mu\downarrow0} \frac{q\mu\phi_1(\mu) - \phi_0(\mu)}{\mu^2} \right\}^{-1/2}$$

$$\overset{(\ddagger)}{=} (q-1)^{1/2} \left\{ \lim_{\mu\downarrow0} \frac{q\phi_1(\mu) + q(q-1)\mu\phi_2(\mu) - q\phi_1(\mu)}{2\mu} \right\}^{-1/2}$$

$$= (q-1)^{1/2} \left\{ \frac{q(q-1)}{2} \lim_{\mu\downarrow0}[\mu^{q-2} + (1-\mu)^{q-2}] \right\}^{-1/2}$$

$$= (q-1)^{1/2} \cdot \left[\frac{q(q-1)}{2}\right]^{-1/2}$$

$$= \sqrt{\frac{2}{q}}.$$

## F.4 Rényi entropy

For the Rényi entropy

$$\phi(\mu) = \frac{1}{q-1} \ln\left[\mu^q + (1-\mu)^q\right],$$

define

$$\phi_0(\mu) = \mu^q + (1-\mu)^q,$$
$$\phi_1(\mu) = \mu^{q-1} - (1-\mu)^{q-1},$$
$$\phi_2(\mu) = \mu^{q-2} + (1-\mu)^{q-2},$$
$$\phi_3(\mu) = \mu^{q-3} - (1-\mu)^{q-3}.$$

When $0 < q < 2$ with $q \neq 1$,

$$\lim_{\mu \downarrow 0} \frac{\phi'(\mu)}{\mu \phi''(\mu)} \left[1 - \frac{\phi(\mu)}{\mu \phi'(\mu)}\right]$$

$$= \lim_{\mu \downarrow 0} \frac{\frac{\phi_1(\mu)}{\phi_0(\mu)}}{(q-1)\mu \frac{\phi_2(\mu)}{\phi_0(\mu)} - q\mu \frac{\phi_1(\mu)^2}{\phi_0(\mu)^2}} \left[1 - \frac{\frac{1}{q-1}\ln\phi_0(\mu)}{\frac{q}{q-1}\mu\frac{\phi_1(\mu)}{\phi_0(\mu)}}\right]$$

$$= \lim_{\mu \downarrow 0} \frac{1}{(q-1)\frac{\mu\phi_2(\mu)}{\phi_1(\mu)} - q\frac{\mu\phi_1(\mu)}{\phi_0(\mu)}} \cdot \left[1 - \frac{\phi_0(\mu)\ln\phi_0(\mu)}{q\mu\phi_1(\mu)}\right]$$

$$= \frac{1}{(q-1)\lim_{\mu\downarrow 0}\frac{\mu\phi_2(\mu)}{\phi_1(\mu)} - q\lim_{\mu\downarrow 0}\frac{\mu\phi_1(\mu)}{\phi_0(\mu)}} \cdot \left[1 - \frac{\lim_{\mu\downarrow 0}\phi_0(\mu)}{q} \cdot \lim_{\mu\downarrow 0}\frac{\ln\phi_0(\mu)}{\mu\phi_1(\mu)}\right]$$

$$= \frac{1}{(q-1)\cdot 1 - q\cdot 0} \cdot \left[1 - \frac{1}{q}\cdot 1\right]$$

$$= \frac{1}{q},$$

where we use

$$\phi_0(\mu) \to 1, \quad \frac{\mu\phi_2(\mu)}{\phi_1(\mu)} = \frac{1 + \left(\frac{\mu}{1-\mu}\right)^{2-q}}{1 - \left(\frac{\mu}{1-\mu}\right)^{2-q}} \to 1, \quad \mu\phi_1(\mu) = \mu^q - \frac{\mu}{(1-\mu)^{1-q}} \to 0,$$

and

$$\frac{\ln\phi_0(\mu)}{\mu\phi_1(\mu)} \overset{(\ddagger)}{\to} \frac{1}{\phi_0(\mu)} \cdot \frac{\phi_0'(\mu)}{\mu\phi_1'(\mu) + \phi_1(\mu)}$$

$$\to \frac{\phi_0'(\mu)}{\mu\phi_1'(\mu) + \phi_1(\mu)}$$

$$= \frac{q\phi_1(\mu)}{(q-1)\mu\phi_2(\mu) + \phi_1(\mu)}$$

$$= \frac{q}{(q-1)\frac{\mu\phi_2(\mu)}{\phi_1(\mu)} + 1}$$

$$\to \frac{q}{(q-1)\cdot 1 + 1}$$

$$= 1.$$

In addition, we have

$$\lim_{\mu \downarrow 0} \frac{\mu}{[\mu\phi'(\mu) - \phi(\mu)]^{1/q}}$$

$$= \left\{\lim_{\mu \downarrow 0} \frac{\mu^q}{\mu\phi'(\mu) - \phi(\mu)}\right\}^{1/q}$$

$$= \left\{ \lim_{\mu \downarrow 0} \frac{(q-1)\mu^q \phi_0(\mu)}{q\mu\phi_1(\mu) - \phi_0(\mu)\ln\phi_0(\mu)} \right\}^{1/q}$$

$$= \left\{ \lim_{\mu \downarrow 0} \frac{(q-1)\mu^q}{q\mu\phi_1(\mu) - \phi_0(\mu)\ln\phi_0(\mu)} \right\}^{1/q} \qquad (\phi_0(\mu) \to 1)$$

$$\overset{(\ddagger)}{=} \left\{ (q-1) \lim_{\mu \downarrow 0} \frac{q\mu^{q-1}}{q\phi_1(\mu) + q(q-1)\mu\phi_2(\mu) - q\phi_1(\mu)\ln\phi_0(\mu) - q\phi_1(\mu)} \right\}^{1/q}$$

$$= \left\{ (q-1) \lim_{\mu \downarrow 0} \frac{\mu^{q-1}}{(q-1)\mu\phi_2(\mu) - \phi_1(\mu)\ln\phi_0(\mu)} \right\}^{1/q}$$

$$\overset{(\ddagger)}{=} \left\{ (q-1) \lim_{\mu \downarrow 0} \frac{(q-1)\mu^{q-2}}{(q-1)\phi_2(\mu) + (q-1)(q-2)\mu\phi_3(\mu) - \frac{q\phi_1(\mu)^2}{\phi_0(\mu)} - (q-1)\phi_2(\mu)\ln\phi_0(\mu)} \right\}^{1/q}$$

$$= \lim_{\mu \downarrow 0} \left\{ \frac{\left[1 + \left(\frac{\mu}{1-\mu}\right)^{2-q}\right] + (q-2)\left[1 - \left(\frac{\mu}{1-\mu}\right)^{3-q}\right] - \frac{q\phi_1(\mu)^2}{(q-1)\mu^{q-2}\phi_0(\mu)} - \left[1 + \left(\frac{\mu}{1-\mu}\right)^{2-q}\right]\ln\phi_0(\mu)}{q-1} \right\}^{-\frac{1}{q}}$$

$$\overset{(A)}{=} \left\{ \frac{1 + (q-2)\cdot 1 - 0 - 1\cdot 0}{q-1} \right\}^{-1/q}$$

$$= 1,$$

where at (A) we used

$$\frac{\phi_1(\mu)^2}{\mu^{q-2}\phi_0(\mu)} \to \mu^{2-q}\phi_1(\mu)^2$$

$$= \mu^q - 2(1-\mu)^{q-1}\mu + (1-\mu)^{2q-2}\mu^{2-q}$$

$$\to -2(1-\mu)^{q-1}\mu + (1-\mu)^{2q-2}\mu^{2-q}$$

$$= \frac{(1-\mu)^{2q-2} - 2(1-\mu)^{q-1}\mu^{q-1}}{\mu^{q-2}}$$

$$\overset{(\ddagger)}{\to} \frac{(2q-2)(1-\mu)^{2q-3} + 2(q-1)(1-\mu)^{q-2}\mu^{q-1} - 2(q-1)(1-\mu)^{q-1}\mu^{q-2}}{(q-2)\mu^{q-3}}$$

$$= \frac{2(q-1)}{q-2} \cdot \frac{1}{(1-\mu)^{2-q}} \cdot \frac{(1-\mu)^{q-1} + \mu^{q-1} - (1-\mu)\mu^{q-2}}{\mu^{q-3}}$$

$$\to \frac{2(q-1)}{q-2} \cdot 1 \cdot \frac{(1-\mu)^{q-1} + \mu^{q-1} - (1-\mu)\mu^{q-2}}{\mu^{q-3}}$$

$$= \frac{2(q-1)}{q-2} \cdot \left[ \left(\frac{\mu}{1-\mu}\right)^{1-q}\mu^2 + \mu^2 - (1-\mu)\mu \right]$$

$$\to 0.$$

When $q = 2$, we leverage

$$\phi_0(\mu) = 2\mu^2 - 2\mu + 1, \quad \phi_1(\mu) = 2\mu - 1, \quad \phi_2(\mu) = 2, \quad \phi_0'(\mu) = 2\phi_1(\mu), \quad \phi_1'(\mu) = 2$$

to have

$$\lim_{\mu \downarrow 0} \frac{\phi'(\mu)}{\mu\phi''(\mu)} \left[ 1 - \frac{\phi(\mu)}{\mu\phi'(\mu)} \right]$$

$$= \lim_{\mu \downarrow 0} \phi_0(\mu) \cdot \frac{2\mu\phi_1(\mu) - \phi_0(\mu)\ln\phi_0(\mu)}{4\mu^2[\phi_0(\mu) - \phi_1(\mu)^2]}$$

$$= \lim_{\mu \downarrow 0} \frac{2\mu\phi_1(\mu) - \phi_0(\mu)\ln\phi_0(\mu)}{4\mu^2[\phi_0(\mu) - \phi_1(\mu)^2]} \qquad (\phi_0(\mu) \to 1)$$

$$\overset{(\ddagger)}{=} \lim_{\mu \downarrow 0} \frac{2[\phi_1(\mu) + 2\mu] - 2\phi_1(\mu)\ln\phi_0(\mu) - 2\phi_1(\mu)}{4\{2\mu[\phi_0(\mu) - \phi_1(\mu)^2] + \mu^2[2\phi_1(\mu) - 4\phi_1(\mu)]\}}$$

$$= \lim_{\mu \downarrow 0} \frac{2\mu - \phi_1(\mu) \ln \phi_0(\mu)}{4\mu[\phi_0(\mu) - \phi_1(\mu)^2 - \mu\phi_1(\mu)]}$$

$$\overset{(\ddagger)}{=} \lim_{\mu \downarrow 0} \frac{2 - 2\ln\phi_0(\mu) - \frac{2\phi_1(\mu)^2}{\phi_0(\mu)}}{4[\phi_0(\mu) - \phi_1(\mu)^2 - \mu\phi_1(\mu)] + 4\mu[2\phi_1(\mu) - 4\phi_1(\mu) - \phi_1(\mu) - 2\mu]}$$

$$= \lim_{\mu \downarrow 0} \frac{1 - \ln\phi_0(\mu) - \frac{\phi_1(\mu)^2}{\phi_0(\mu)}}{2[\phi_0(\mu) - \phi_1(\mu)^2 - 4\mu\phi_1(\mu) - 2\mu^2]}$$

$$\overset{(\ddagger)}{=} \lim_{\mu \downarrow 0} \frac{-\frac{2\phi_1(\mu)}{\phi_0(\mu)} - \frac{4\phi_0(\mu)\phi_1(\mu) - 2\phi_1(\mu)^3}{\phi_0(\mu)^2}}{2[2\phi_1(\mu) - 4\phi_1(\mu) - 4\phi_1(\mu) - 8\mu - 4\mu]}$$

$$= \lim_{\mu \downarrow 0} \frac{\phi_1(\mu)}{\phi_0(\mu)^2} \cdot \frac{3\phi_0(\mu) - \phi_1(\mu)^2}{6[\phi_1(\mu) + 2\mu]}$$

$$= \frac{1}{3}.$$

In addition, defining

$$\zeta := \frac{2\mu\phi_1(\mu) - \phi_0(\mu)\ln\phi_0(\mu)}{\phi_0(\mu)},$$

we have

$$(\xi :=) \lim_{\mu \downarrow 0} \frac{\mu}{[\mu\phi'(\mu) - \phi(\mu)]^{1/3}}$$

$$= \lim_{\mu \downarrow 0} \frac{\mu}{\left[\frac{2\mu\phi_1(\mu) - \phi_0(\mu)\ln\phi_0(\mu)}{\phi_0(\mu)}\right]^{1/3}} \quad \left(\text{implies } \xi = \lim_{\mu \downarrow 0} \mu\zeta^{-1/3}; \text{ we will use this below at (\$)}\right)$$

$$\overset{(\ddagger)}{=} \lim_{\mu \downarrow 0} \frac{3\zeta^{2/3}}{\frac{2\phi_1(\mu)}{\phi_0(\mu)} + 2\mu\frac{2\phi_0(\mu) - 2\phi_1(\mu)^2}{\phi_0(\mu)^2} - \frac{2\phi_1(\mu)}{\phi_0(\mu)}}$$

$$= \lim_{\mu \downarrow 0} \frac{3\phi_0(\mu)^2}{4} \frac{\zeta^{2/3}}{\mu[\phi_0(\mu) - \phi_1(\mu)^2]}$$

$$\overset{(\ddagger)}{=} \lim_{\mu \downarrow 0} \frac{3}{4} \frac{\frac{2}{3} \cdot 2\mu\frac{2\phi_0(\mu) - 2\phi_1(\mu)^2}{\phi_0(\mu)^2}}{\{\phi_0(\mu) - \phi_1(\mu)^2 + \mu[2\phi_1(\mu) - 4\phi_1(\mu)]\}\,\zeta^{1/3}}$$

$$= \lim_{\mu \downarrow 0} \frac{2\mu[\phi_0(\mu) - \phi_1(\mu)^2]}{[\phi_0(\mu) - \phi_1(\mu)^2 - 2\mu\phi_1(\mu)]\zeta^{1/3}}$$

$$\overset{(\ddagger)}{=} \lim_{\mu \downarrow 0} \frac{3\zeta^{2/3}}{2} \frac{1}{\mu\phi_0(\mu)[\phi_0(\mu) - \phi_1(\mu)^2] - 3\frac{\phi_0(\mu)^3[\phi_1(\mu) + \mu]\zeta}{\phi_0(\mu) - \phi_1(\mu)^2 - 2\mu\phi_1(\mu)}}$$

$$= \lim_{\mu \downarrow 0} \frac{3\zeta^{2/3}}{2} \frac{1}{\mu[\phi_0(\mu) - \phi_1(\mu)^2] + \frac{3\zeta}{\phi_0(\mu) - \phi_1(\mu)^2 - 2\mu\phi_1(\mu)}}$$

$$= \left\{\lim_{\mu \downarrow 0} \frac{2}{3}\frac{\phi_0(\mu) - \phi_1(\mu)^2}{\mu} \cdot (\mu\zeta^{-1/3})^2 + \lim_{\mu \downarrow 0} \frac{2\mu}{\phi_0(\mu) - \phi_1(\mu)^2 - 2\mu\phi_1(\mu)}\frac{1}{\mu\zeta^{-1/3}}\right\}^{-1}$$

$$\overset{(\$)}{=} \left\{\frac{2}{3}\xi^2 \lim_{\mu \downarrow 0}(2 - 2\mu) + \frac{1}{\xi}\lim_{\mu \downarrow 0}\frac{1}{2 - 3\mu}\right\}^{-1}$$

$$= \left\{\frac{4}{3}\xi^2 + \frac{1}{2\xi}\right\}^{-1},$$

which implies

$$\xi = \frac{1}{\frac{4}{3}\xi^2 + \frac{1}{2\xi}}.$$

By solving this, we have

$$\lim_{\mu \downarrow 0} \frac{\mu}{[\mu\phi'(\mu) - \phi(\mu)]^{1/3}} = \xi = \left(\frac{3}{8}\right)^{1/3}.$$

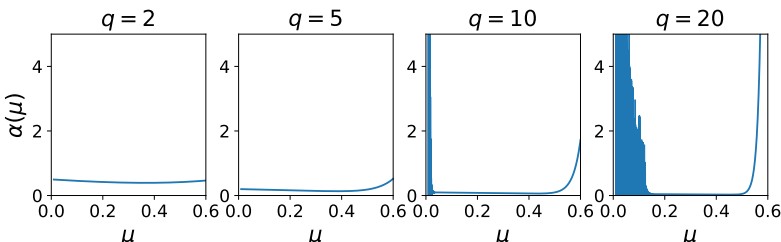

**Figure 3:** For the pseudo-spherical entropy, $\alpha(\mu) = [\phi'(\mu)/\mu\phi''(\mu)] \cdot [1 - \phi(\mu)/\mu\phi'(\mu)]$ is shown.

### F.5 Pseudo-spherical entropy

Consider the $q$-pseudo-spherical entropy $\phi(\mu) = [\mu^q + (1 - \mu)^q]^{1/q} - 1$ for $q > 1$ [24]. It is also known as the $q$-norm (neg)entropy [12]. When $q = 2$, it recovers the spherical entropy associated with the spherical loss [1]. When $q \uparrow \infty$, it approaches $\phi_\infty(\mu) = \max\{\mu, 1 - \mu\} - 1$, which is the Bayes risk of the hinge/0-1 losses [15]. As seen in Figure 3, the limit $\alpha$ (in (3)) does not exist, which indicates that we cannot guarantee the $\varepsilon$-optimal risk for vanishingly small $\varepsilon$.

