# OpenReview forum: "Any-stepsize Gradient Descent for Separable Data under Fenchel–Young Losses"
_NeurIPS.cc/2025/Conference — NeurIPS 2025 spotlight_

### Official Review · Reviewer_Trfc · 2025-06-10

**Clarity:** 3
**Significance:** 3
**Originality:** 3
**Rating:** 5
**Confidence:** 4

**Summary:**

This paper investigates gradient descent in the edge of stability (EoS) where step sizes are chosen to be much larger than the classical stable regime $1/\beta$ for minimization of $\beta$-smooth functions. The main contribution in this work is in extending the analysis of large step size GD for losses that satisfy the self bound property (e.g., logistic) to Fenchel-Young losses (Bregman divergences). In particular, this work shows that large step sizes achieve accelerated convergence for some loss functions that violate the self bounding property.

**Questions:**

* Since the authors consider a variant of Bregman divergence, a natural algorithm is to use mirror descent. In particular, GD is sub-optimal in the dimension dependence. Can GD outperform mirror descent in high dimensions using large step sizes?

**Ethical Concerns:**

["NO or VERY MINOR ethics concerns only"]

**Final Justification:**

I have raised my score from 4 to 5. My recommendation is based on the paper's solid contribution and the authors' constructive engagement.

- The paper's presentation issues were addressed appropriately. My initial concerns about presentation clarity have been effectively addressed. The authors proposed a specific re-phrasing that I agree is much clearer. I believe that by incorporating this, along with the excellent presentation suggestions from Reviewer 1, the final camera-ready version will be significantly improved.
- Scope: My main concern was its extension to the stochastic setting. The authors addressed this concern, and I am encouraged by their response. However, because the detailed analysis for the stochastic case is not included in the submitted manuscript, it is difficult to fully assess the significance and correctness of this important extension.

Given the strength of the work and the resolution of my concerns, I am happy to recommend this paper for acceptance.

**Limitations:**

The authors clearly state a few limitations in Section 5.

**Paper Formatting Concerns:**

No major formatting issues have been found.

**Quality:**

3

**Strengths And Weaknesses:**

**Strengths**

* This work shows that large stepsize GD converges at an accelerated rate for a larger class of loss functions that was known previously.

* The generality of the Fenchel-Young losses is used to compare the rate achieved by large step size GD compared to the rate achieved by classical (small step size) GD. As shown in Table 1, large step sizes accelerate convergence for Tsallis and Renyi entropies, but not for Shannon or Semi-circle losses.

**Weaknesses**

* Confusing wording: Authors use the wording “arbitrary” step size, referring to how it is independent of the error tolerance. This choice of wording makes it hard to follow the paper, since this is used in claims such as “GD converges with arbitrary step size” throughout the paper. This is not true, since GD remains in the EoS phase for O(step size) iterations.

* While the results are interesting, the claim that achieving the accelerated rate is “groundbreaking” seems like an overstatement, given that the EoS phenomenon has been observed in prior works.

* The authors consider full-batch GD and prove convergence of the training error. As discussed in Section 5, all points are therefore classified in the EoS phase. When aiming for perfect training accuracy, it’s unclear whether large step sizes are truly useful.

* It would be more interesting to see if large step sizes achieve accelerated convergence in a population error under the stochastic setting. [Wu et al. 2024] show that this is not the case for the logistic loss or 0-1 loss. Can accelerated convergence in the population error be shown for the Fenchel-Young losses considered in this work? Without any practical benefits to using large step sizes, the motivation for studying this problem is unclear.

---

> ### Author Rebuttal · Authors · 2025-07-25
>
> We highly appreciate your critical feedback. We will revise wordings based on your suggestions.
>
> **W1 (“arbitrary” stepsize).** We understand that you have a concern that we should not say “GD converges” if it stays in EoS. How about rephrasing this as “large-stepsize GD attains an $\\epsilon$-optimal solution”?
>
> **W3 (benefit of large stepsize).** We basically agree with the reviewer. When we are only interested in minimizing the training risk, the convergence rate provided in Theorem 5 does not improve with large $\\eta$. Therefore, we did not emphasize the benefit of large stepsize intentionally, but instead focus on the mechanism of convergence beyond the classical threshold $\\eta\\leq2/\\beta$.
>
> **W4 (SGD).** In short, our GD convergence analysis can be directly extended to SGD case, and the convergence rate remains the same, thanks to the perceptron structure.
>
> To consider the stochastic scenario, let’s suppose the same assumption as Wu et al. (2024, Assumption 2). The parameter is updated by $\\mathbf{w}\_{t+1}=\\mathbf{w}\_t - \\eta\\nabla L\_t(\\mathbf{w}\_t)$ with $L\_t(\\mathbf{w}) = \\ell(\\langle y_t\\mathbf{x}\_t, \\mathbf{w}\\rangle)$ ($\\ell$: Fenchel–Young loss), where a single data $(\\mathbf{x}\_t,y\_t)$ is sampled i.i.d. based on the following assumptions:
>
> + $\\|\\mathbf{x}\\| \\leq 1$ almost surely
> + there exist a margin $\\gamma > 0$ and a unit vector $\\mathbf{w}\_\*$ such that $\\langle y\\mathbf{x},\\mathbf{w}\_\*\\rangle \\geq \\gamma$ almost surely
>
> The main idea of our proof is Eq. (4). The upper bound of $\\langle\\mathbf{w}\_t,\\mathbf{w}\_\*\\rangle$ can be shown by the split optimization technique (Eq. (5)), which can be extended to the stochastic scenario as shown in Wu et al. (2024, Lemmas 16 and 17). The lower bound of $\\langle\\mathbf{w}\_t,\\mathbf{w}\_\*\\rangle$ at l.214 also holds (almost surely) straightforwardly. Hence, our proof is almost directly extendable to the stochastic scenario as long as we suppose the above assumptions.
>
> This point seems to be interesting in contrast to Wu et al. (2024). Their convergence rate is not accelerated for SGD because the modified descent lemma holds only for the deterministic case and the convergence radius is limited to $\\eta\\lesssim T$. In contrast, our convergence seems to hold for SGD and the convergence radius is not limited.
>
> **Q (mirror descent).** Interesting question. Can you provide a reference where mirror descent has better dimension dependence than GD? One challenge to study this question is that we don’t have an explicit dimension dependence in Theorem 5. The dimension-related factor is hidden in the margin $\\gamma$ instead. So we should investigate whether mirror descent improves the dependency in $\\gamma$.

---

> ### Comment · Reviewer_Trfc · 2025-08-03
>
> Thank you for the detailed response. This is a strong paper with a valuable contribution, and I recommend accepting the paper.
>
> I appreciate your interesting response regarding the extension to the stochastic setting, and it is encouraging to hear that the results hold. I strongly encourage the authors to include the full analysis in the camera-ready version if possible, as it would significantly strengthen the paper.
>
> In response to the wording, yes, I believe the rephrasing is much clearer.

---

### Official Review · Reviewer_7Hzu · 2025-07-02

**Clarity:** 4
**Significance:** 3
**Originality:** 3
**Rating:** 5
**Confidence:** 4

**Summary:**

This paper investigates gradient descent (GD) with arbitrary stepsizes in the setting of binary classification with linearly separable data, focusing on a broad class of loss functions called Fenchel–Young losses. The work extends prior results that showed convergence for logistic loss under large stepsizes, notably beyond the classical “edge of stability” (EoS) regime where standard convergence guarantees no longer apply.

**Questions:**

1. **On the dependence on the number of samples $n$ in Theorem 5:**
The theorem’s bound scales with $n^{1+\alpha}$, but intuitively, shouldn't the convergence rate be independent of $n$? For instance, duplicating each sample (so $n \to 2n$) does not affect the empirical risk $L(w)$, yet the bound would change. Could the authors clarify this apparent inconsistency?

2. **On the tightness of the bound in Theorem 5:**
Can you comment on how tight your convergence bound is, at least empirically? For example, for the **Shannon entropy** (i.e., logistic loss), it seems from Figure 1 that **doubling the stepsize approximately doubles convergence speed**, as expected by your theoretical predictions?  Can other such comments be made?
Also, when the stepsize $\eta$ is very large, the theoretical convergence rate saturates at $\mathcal{O}(n^2 / (\gamma^2 \ln 2 \cdot t))$. Can you comment on whether this behaviour is observed experimentally?

3. **On arbitrarily large stepsizes:**
Your results suggest that using the larger the stepsize, the better the convergence rate. But is there any practical downside (e.g., stability, generalisation) to this claim? How is this compatible whith the EoS perspective where GD is initially in an oscillation phase: shouldn't this oscillation phase be arbitrarily long when taking $\eta$ to $\infty$?

4. **On the slow convergence for Rényi entropy with $q < 1$:**
For Rényi entropy with $q = 0.5$, Theorem 5 yields a **slow convergence rate** of $\Omega(\varepsilon^{-4})$, which seems to align with what is observed in Figure 1. However, **gradient descent on smooth convex losses** typically achieves $\mathcal{O}(1/\varepsilon)$ convergence. What prevents this from holding here? Is the associated loss function **not $L$-smooth**, or is some other property lacking?

**Ethical Concerns:**

["NO or VERY MINOR ethics concerns only"]

**Final Justification:**

The paper is well written and the theoretical contributions are clear and justified, which is why I recommend the acceptance of this paper.

**Quality:**

4

**Strengths And Weaknesses:**

**Strengths**
This paper is very well written and easy to follow, even for readers who are not familiar with Fenchel–Young losses. The presentation is well structured, with key ideas introduced in a logical and intuitive progression. The theoretical contributions are clearly stated, carefully justified, and supported by concrete examples that help illustrate and clarify the main results. Finally, while the experiments are minimal, they provide clear and effective illustrations that support the theoretical claims.

**Weaknesses**
The focus on Fenchel–Young losses, while theoretically appealing, may limit the practical relevance of the work. In practice, most machine learning applications rely on standard losses such as the logistic or hinge loss. The motivation for exploring this broader class—beyond unification or analytical convenience—is not entirely clear.
The paper would benefit from a more detailed discussion of practical scenarios in which non-standard Fenchel–Young losses might be preferable. For example, are there settings (e.g., structured prediction, robustness to noise, or implicit regularisation effects) where the improved convergence rates offered by certain losses lead to practical advantages?

---

> ### Author Rebuttal · Authors · 2025-07-25
>
> We appreciate the reviewer for taking the time to provide feedback.
>
> **W (practicality of FY loss).** Although the logistic or hinge losses are widely prevailing in practice, there is room to improve the practical performance by changing loss functions. Notably, Roulet et al. (2025) reported that the Tsallis loss occasionally outperforms the logistic loss even with the simplest ImageNet classification. Our study would be relevant to its mechanism since one of the biggest differences between the logistic and Tsallis losses is separation margin.
>
> Roulet et al. (2025) “Loss Functions and Operators Generated by f-Divergences”
>
> **Q1 (dependence on n).** In theory, the dependency of the rate on $n$ is due to the worst-case analysis at l.572–575: for example, $\\tfrac1n\\sum\_{j\\in[n]} g(\\langle\\mathbf{w}\_k,\\mathbf{z}\_j\\rangle) \\geq \\tfrac1ng(\\langle\\mathbf{w}\_k,\\mathbf{z}\_i\\rangle)$ for some worst data point $i$. The $\\tfrac1n$ factor eventually leads to the $n$-dependency. This means there always exists a bad data point $i$ with a negatively large prediction margin $\\langle\\mathbf{w}\_k,\\mathbf{z}\_i\\rangle$ before achieving $\\epsilon$-optimal loss.
>
> Such a worst-case is unlikely because most data points tend to cluster in similar directions, and we conjecture that the $n$-dependency can be improved with an extra assumption. Actually, the scenario of duplicated samples suggested by the reviewer falls in this situation.
>
> In practice, Tyurin (2025, Section 6) eliminates the $n$-dependency by adding a multiplicative adaptive factor to stepsize. A similar approach could work for Fenchel–Young losses.
>
> Tyurin (2025) “From Logistic Regression to the Perceptron Algorithm: Exploring Gradient Descent with Large Step Sizes”
>
> **Q2 (tightness).** In short, we recognize our analysis is not tight enough.
>
> + _Shannon case:_ Since the corresponding logistic loss has the self-bounding property, GD experiences the phase transition, shown by Wu et al. (2024). The doubled convergence speed by the doubled stepsize should be explained by the convergence rate after the phase transition, not by our Theorem 5, because Theorem 5 characterizes the convergence rate only before the phase transition (discussed implicitly at l.314–316; we will clarify more).
> + _Tsallis case:_ Since the corresponding loss function has separation margin (for Tsallis-1.5 and -2.0), GD does not experience the phase transition. When $q=2.0$, Theorem 5 gives $\epsilon=O(1/T^2)$. Actually, the trajectory in Figure 1 is faster than this rate, and close to the exponential rate. We conjecture that an extra assumption on the dataset is necessary to obtain the near-exponential rate just because of the similar reason to Q1.
>
> **Q3 (downside of large stepsize).**
>
> + _For a loss function with separation margin:_ We do not see downsides of arbitrarily large stepsize so far. To make it sure, Theorem 5 characterizes the convergence rate in EoS phase only. If we look at Tsallis-1.5 and -2.0 in Figure 1, the convergence is actually faster with larger stepsize, and they stay on EoS. For example, Tsallis-2.0 with $\\eta=2^1$ apparently decreases with increasing $t$, but the loss value keeps oscillating locally.
> + _For a self-bounding loss function:_ As the reviewer noted, EoS phase continues longer with larger stepsize. This is detrimental to achieving the accelerated convergence rate obtained by Wu et al. (2024).
>
> **Q4 (slower convergence than $O(1/\\epsilon)$).** The typical GD convergence rate $O(1/\\epsilon)$ for smooth convex losses is valid only for small enough stepsize $\\eta\\le2/\\beta$. The stepsize used in Figure 1 is larger than this threshold, which prevents from achieving $O(1/\\epsilon)$. Note that we plotted Tsallis only in Figure 1.

---

> > ### Comment · Reviewer_7Hzu · 2025-08-06
> >
> > I thank the authors for their responses to my questions.
> >
> > As mentioned in my original review, the paper is well written, and the contributions are clearly stated and well justified, which supports my rating of 5/6. However, I remain unconvinced about the strong significance and potential impact of the work, and for this reason, I will maintain the score.

---

### Official Review · Reviewer_i177 · 2025-07-03

**Clarity:** 3
**Significance:** 3
**Originality:** 3
**Rating:** 5
**Confidence:** 3

**Summary:**

This paper studies GD convergence under arbitrarily large stepsize. Previous studies have noticed that certain objective functions still converge when optimized by GD, even with excessively large step sizes. Researchers have been trying to understand what properties of functions allow this to happen, and this paper adds to this line of research. While earlier work showed that functions with a self-bounding property can make GD converge with any step size, this paper extends the result by demonstrating that the self-bounding property isn't necessary. Specifically, it proves that GD with any step size converges on Fenchel-Young losses with a separable margin, which may not have the self-bounding property. Moreover, this paper instantiates the Fenchel-Young losses with certain examples, including Shannon entropy, full circle entropy, Tsallis entropy, and Renyi entropy, and provided explicit convergence rates for each loss. Notably, Tsallis q-loss achieves $O(\epsilon^{-1/2})$ for $q>2$ and Renyi 2-entropy achieves $O(\epsilon^{-1/3})$, both faster than the standard $O(\epsilon^{-1})$ rate of GD on unstructured convex and smooth losses.

**Questions:**

NA

**Ethical Concerns:**

["NO or VERY MINOR ethics concerns only"]

**Final Justification:**

Sorry for the late response, I wasn't aware of the new requirement of this year. I have read the responses from the authors and the comments from other reviewers. I keep my current score as the recommended score.

**Limitations:**

yes

**Quality:**

3

**Strengths And Weaknesses:**

This paper contributes to the literature by extending the line of research on GD convergence under arbitrarily large stepsize. Previous studies focused on functions with a self-bounding property, and this paper broadens the family of objective functions by demonstrating that the self-bounding property is not necessary. Instead, it shows that GD converges on Fenchel-Young losses with separable margins, which may not possess the self-bounding property. This extension enhances the understanding of what properties of loss functions allow convergence with large stepsize.

Moreover, it is notable that certain instantiation of Fenchel-Young loss, such as Tsallis q-loss and Renyi q-loss for $q>1$ achieve a faster rate compared to the standard $O(\epsilon^{-1})$ rate. Such result is also new to the field.

In addition, I find the proof technique of this paper inspiring. Instead of standard analysis that build on some variants of descent lemma, this paper uses the perceptron method. This technique could inspire further research and exploration.

As a limitation, the current analysis is restricted to the binary classification problem, where the loss function can be reduced to a Fenchel-Young loss on 1d prediction namely $\phi^*(-z)$, and a linear model where the prediction can be formulated as $\langle w, x\rangle$. Extending to a more complicatedly structured model such as deep neural networks will be an interesting (and definitely very challenging) result.

---

> ### Author Rebuttal · Authors · 2025-07-25
>
> We would like to thank the reviewer for finding our result and proof techniques interesting. On the limitations, while extending to the multi-class case may not be straightforward, we guess it’s elementary to extend the linear model to deep homogeneous networks as done in several papers such as Tsilivis et al. (2025) because deep homogeneous networks can be analyzed similarly to linear models.
>
> Tsilivis et al. (2025) “Flavors of Margin: Implicit Bias of Steepest Descent in
> Homogeneous Neural Networks”

---

> > ### Comment · Reviewer_i177 · 2025-08-07
> >
> > I thank the authors for their response and pointing to this reference. I don't have further questions and remain my current score.

---

### Official Review · Reviewer_Fpmo · 2025-07-03

**Clarity:** 3
**Significance:** 3
**Originality:** 4
**Rating:** 5
**Confidence:** 4

**Summary:**

This work sits at the intersection of two major threads in optimization for machine learning:

### **1. Convergence Analyses for Specific Loss Functions**

One line of work builds on the success of convex optimization theory to develop fine-grained convergence analyses tailored to particular loss functions—such as logistic regression—that arise frequently in machine learning. The goal is to go beyond general smooth convex convergence rates and exploit structural properties (e.g., higher-order smoothness) to obtain faster rates or design better algorithms. A representative example is the [self-concordant analysis of logistic regression by Francis Bach](https://www.di.ens.fr/~fbach/bach_ejs_self_concordance.pdf), along with several follow-up works. These efforts aim to isolate specific regularity assumptions that make certain loss functions easier to optimize than general convex functions. The broader hope is to both explain faster convergence and to inspire new algorithms that leverage loss-specific structure.

### **2. Edge of Stability (EoS)**

A separate thread of work focuses on phenomena observed in deep learning, particularly the so-called *edge-of-stability* (EoS). Empirically, training with a near-optimal step size often leads to dynamics where the model navigates increasingly sharper regions of the loss landscape, eventually stabilizing at a level of sharpness dictated by the step size. Intriguingly, step sizes that work well in practice are often significantly larger than those suggested by classical smoothness-based bounds (e.g., the $1/\beta$ rule from the descent lemma).

This raises a natural question: Are the classical step-size constraints overly conservative? And if so, for which functions can we prove convergence with larger—or even arbitrary—step sizes, and what are the resulting rates?

### **Wu et al. and Logistic Regression**

At the intersection of these two threads lies the work of Wu et al., which revisits logistic regression but does not exclusively use classical descent-based analysis and higher-order smoothness arguments. Instead, they provide a phase-based analysis of gradient descent, identifying two distinct phases: an initial non-monotonic phase where the loss fluctuates but decreases on average, and a second phase of accelerated convergence. This allows them to prove a $1/\sqrt{\epsilon}$ convergence rate—faster than the standard $1/\epsilon$ rate for non-accelerated methods.

Interestingly, this perspective also offers insight into EoS: perhaps the phenomenon is not unique to deep networks, but rather arises from commonly used loss functions, such as logistic loss, which can exhibit EoS-type behavior even in simpler settings. That said, Wu et al. do not discount the role of representation learning and multiple phases in deep models as contributing factors to EoS. Subsequent work has extended their analysis to shallow homogeneous networks.

### **This Paper**

This paper contributes to both lines of research while offering a fresh perspective on the work of Wu et al. Rather than relying on higher-order smoothness or phase-based arguments, the analysis is inspired by the classical perceptron algorithm. It demonstrates that for a broad class of loss functions—specifically, Fenchel-Young losses derived from suitable potential functions—faster convergence is possible under a "separation margin" condition.

Fenchel-Young losses are convex and often exhibit desirable properties as surrogate losses, such as better calibration depending on the choice of potential. While the paper does not aim to characterize which potentials are most beneficial from a learning-theoretic perspective, it leverages this class to showcase how convergence rates can improve with a separation margin.

The paper (along with some previous work) also argues that while the two-phase analysis of Wu et al. offers valuable insight, it is fundamentally tailored to logistic regression. And when the data is separable, the second phase in Wu et al. is less relevant for the downstream classification task. Notably, the paper demonstrates that logistic regression, despite possessing self-bounding gradients, fails to satisfy a separation-margin condition and therefore cannot benefit from the improved convergence guarantees provided by the new analysis. As a result, the perceptron-based analysis can not beat the classical $O(1/\epsilon)$ rate. This raises the question: Does the accelerated convergence in the later phase of Wu et al.'s analysis "truly capture" why EoS is a valuable phase in deep learning tasks? While the paper does not answer this question, it suggests the answer might be no. In particular, the paper highlights that by optimizing losses with a separation margin that enables faster optimization, binary classification on separable data can be performed more efficiently, leaving room for future inquiry into these loss functions.

Finally, the paper supports its theoretical findings with experiments, illustrating faster convergence—and convergence with arbitrary step sizes—for select Fenchel-Young losses that admit a separation margin.

**Questions:**

I have suggested making certain changes in the weakness above, but these are minor organization changes and won't change my score. They would make the paper more readable though. I have some questions about the results and the extensions:
1. Does the perceptron based view, extend more easily to the non-separable case? The analysis of Wu et al. seems particularly hard to extend to the case of non-seperable data, and can thus arguably not capture many real world artefacts/noise in the data. If not, can the authors highlight the key difficulty.
2. Can the perceptron based view, accommodate noise? In general, it would be valuable to address this, as the edge-of-stability phenomenon can also arise with stochastic gradient methods and their variants.
3. Are there any other natural loss function classes where the authors can extend similar ideas?

**Ethical Concerns:**

["NO or VERY MINOR ethics concerns only"]

**Final Justification:**

I am happy with the author's response. I don't have any major outstanding concerns, and as such recommend accepting the paper.

**Limitations:**

While there are no major limitations, it may be helpful to comment on the extent to which the current analysis extends to loss functions without bounded gradients, particularly in cases where the functions exhibit higher-order smoothness. More broadly, the related work section could be strengthened by including discussions of higher-order smoothness assumptions—such as quasi-self-concordance—which are notably absent. Additionally, the paper only studies exact gradient descent.

**Quality:**

3

**Strengths And Weaknesses:**

The Perceptron algorithm and the analysis of its mistake bound are classic theoretical results, often taught early in foundational machine learning courses. Demonstrating that this analysis extends to a much more general setting—namely, to Fenchel-Young losses—has both scientific and pedagogical value. This alone provides sufficient justification for accepting the paper. Furthermore, the paper demonstrates that this perspective can enhance convergence rates for specific losses, both in terms of speed and robustness to step-size choices. Overall, the paper is technically solid and clearly contextualized within the relevant literature.

That said, the writing is lacking in many places, with frequent grammatical errors and awkward phrasings. The entire related work section is riddled with such issues—too many to list exhaustively. I strongly recommend a careful revision of this section, ideally using a grammar checker or editor. Beyond Section 1.1, the following lines contain grammatical mistakes or awkward constructions: 125–127, 135, 224, 225, 245, 311, 329, 330.

Some key definitions, such as that of the *separation margin*, appear only later in the paper, despite being referenced early in the introduction. This creates confusion for the reader—for instance, it is unclear what it means for a "loss to entail a separation margin." Definitions critical to understanding the main contributions should be presented earlier or clearly foreshadowed.

The background discussion in Section 2 also feels rushed and lacks sufficient context. If the paper is accepted, the authors should consider using the additional page to expand the preliminaries. Specifically, it would be helpful to clearly address the following: *"Loss X has desirable properties as a surrogate for downstream task D, and our analysis shows faster optimization for Loss X—therefore, we suggest a better algorithm for the downstream task D."* This line of reasoning is currently implied but not clearly articulated. While the authors note in Section 5 that losses with a separation margin are suitable for classification, the connection between surrogate loss properties and optimization guarantees can be made more explicit.

In fact, the discussion around surrogate loss selection and what the authors refer to as "regret with respect to a downstream task" (lines 110–113) is not entirely beyond the scope of the work. For example, logistic regression is a widely used surrogate for classification due to its theoretical guarantees and empirical performance. This justifies the need to study its optimization behavior in more detail, even if it does not admit a separation margin.

I also recommend creating a separate section titled **"Comparison with [59]"** and moving the relevant discussion from the current "Phase Transition" subsection there. It is essential to clearly explain how the results in this paper relate to those in [59]. Because logistic regression lacks the separation margin property, the perceptron-style analysis in this paper cannot yield better than $1/\epsilon$ rates, while [59] achieves $1/\sqrt{\epsilon}$ via a phase-wise argument. As mentioned in the summary, the current paper suggests that the phase-wise view might offer limited insight for separable classification problems, where the optimization target is achieved early. Section 5 briefly touches on how phase transitions might have auxiliary benefits, but these points are not clearly or explicitly presented. Improving the organization and writing of this section would strengthen the paper.

Finally, it would be valuable to elaborate on how perceptron-style analysis might yield useful insights for optimizing deep networks, and what role the separation margin plays in that broader context. While the paper leaves these as open questions, they are arguably the most compelling directions beyond the optimization theory itself.

---

> ### Author Rebuttal · Authors · 2025-07-25
>
> Thank you for the time and effort. We will fix the grammatical errors and reorganize the paragraphs based on your feedback. Subsequently, we mainly focus on answering your questions.
>
> **W1 (downstream regret).** Upon the acceptance, we will expand Section 2 with an extra page to describe the preliminary of the connection between a surrogate loss and a downstream task.
> We initially said "regret with respect to a downstream task" (lines 110–113) is beyond our scope because we pay more attention to the optimization aspect. The study of a downstream task performance is certainly relevant, so we will remove the phrase “beyond our scope.”
>
> **Q1 (non-separable data).** Our proof relies on the recursive expansion of $\\langle\\mathbf{w}\_t,\\mathbf{w}\_\*\\rangle$ (l.214), where the linear separability is used. Note that the linear separability implies $\\langle\\mathbf{w}\_\*,\\mathbf{z}\_{i\\_t-1}\\rangle \\geq \\gamma$, which yields the first lower bound. Without the linear separability, this recursive step becomes infeasible. We may introduce some additional assumptions on the data distribution to rescue this step, which seems an interesting open question.
>
> **Q2 (SGD).** In short, our GD convergence analysis can be directly extended to SGD case, and the convergence rate remains the same, thanks to the perceptron structure.
>
> To consider the stochastic scenario, let’s suppose the same assumption as Wu et al. (2024, Assumption 2). The parameter is updated by $\\mathbf{w}\_{t+1}=\\mathbf{w}\_t - \\eta\\nabla L\_t(\\mathbf{w}\_t)$ with $L\_t(\\mathbf{w}) = \\ell(\\langle y_t\\mathbf{x}\_t, \\mathbf{w}\\rangle)$ ($\\ell$: Fenchel–Young loss), where a single data $(\\mathbf{x}\_t,y\_t)$ is sampled i.i.d. based on the following assumptions:
>
> + $\\|\\mathbf{x}\\| \\leq 1$ almost surely
> + there exist a margin $\\gamma > 0$ and a unit vector $\\mathbf{w}\_\*$ such that $\\langle y\\mathbf{x},\\mathbf{w}\_\*\\rangle \\geq \\gamma$ almost surely
>
> The main idea of our proof is Eq. (4). The upper bound of $\\langle\\mathbf{w}\_t,\\mathbf{w}\_\*\\rangle$ can be shown by the split optimization technique (Eq. (5)), which can be extended to the stochastic scenario as shown in Wu et al. (2024, Lemmas 16 and 17). The lower bound of $\\langle\\mathbf{w}\_t,\\mathbf{w}\_\*\\rangle$ at l.214 also holds (almost surely) straightforwardly. Hence, our proof is almost directly extendable to the stochastic scenario as long as we suppose the above assumptions.
>
> This point seems to be interesting in contrast to Wu et al. (2024) because their convergence proof based on the modified descent lemma holds only for the deterministic case, but our perceptron-based proof remains valid in the stochastic case.
>
> **Q3 (other natural loss functions).** The general idea of our convergence proof lies in Eq. (4), for which we need to estimate the constants $C\_L$ and $C\_U$. For other loss functions beyond Fenchel–Young losses, $C\_U$ can be estimated without many challenges via Eq. (5), while estimating $C\_L$ needs a bit more technicality. This requires a lower bound on the loss derivative $g$ (l.578–579), but we are not sure for now whether a similar polynomial bound like $g(\\epsilon) \\gtrsim \\epsilon^{\\alpha}$ can be obtained for broader loss functions.

---

> > ### Comment · Reviewer_Fpmo · 2025-08-03
> >
> > Thank you for the detailed comments addressing my questions. I think the paper will undoubtedly improve with the writing changes.
> >
> > I will retain my score and recommend accepting the paper.

---

### Decision · Program_Chairs · 2025-09-17

**Decision:**

Accept (spotlight)

**Comment:**

**Summary:** Previous studies have shown that for certain objective functions (e.g. logistic regression), GD with large stepsizes still converges. While earlier work showed that functions with a self-bounding property can make GD converge with any step size, this paper extends the result by demonstrating that the self-bounding property is not necessary. Specifically, it proves that GD with any stepsize converges on Fenchel-Young losses with a separable margin, which may not have the self-bounding property. Moreover, this paper instantiates the Fenchel-Young losses with certain examples, including Shannon entropy, full circle entropy, Tsallis entropy, and Renyi entropy, and provides explicit convergence rates for each loss.

**Strengths:**
- Going beyond the exponential/logistic losses systematically to consider Fenchel-Young losses is novel and interesting
- Nice proof technique that builds on the perceptron method, and can easily handle stochasticity
- Well-written paper with a clear contribution, and cleanly presented theoretical results.

**Weaknesses:**
- Lack of examples justifying the importance and use of Fenchel Young losses.

**Decision and Suggested Changes:** The reviewers unanimously agree that the paper is well-written, and its contributions merit acceptance. After carefully reading the rebuttal/discussion, I tend to agree. Please incorporate the reviewer comments in the final version of the paper. In particular, addressing the following concerns will help strengthen the current version of the paper:
- Discussion about the non-separability and extension to SGD (response to Rev. Fpmo, Rev. Trfc)
- Discussion why extending to the multi-class classification setting is non-trivial (Rev. i177)
- Discussion about the dependence on $n$ and the tightness of the bound (Rev. 7Hzu)
- Incorporating the presentation suggestions from Rev. Fpmo